# Impact of 3D Cloud Structures on the Atmospheric Trace Gas Products from UV-VIS Sounders - Part III: bias estimate using synthetic and observational data

Arve Kylling[1], Claudia Emde[2], Huan Yu[3], Michel van Roozendael[3], Kerstin Stebel[1], Ben Veihelmann[4], and Bernhard Mayer[2]

[1]NILU - Norwegian Institute for Air Research, Kjeller, Norway
[2]Ludwig-Maximilians-University, Meteorological Institute, Munich, Germany
[3]Belgian Institute for Space Aeronomy, Brussels, Belgium
[4]ESA-ESTEC, Noordwijk, the Netherlands

*Correspondence to:* Arve Kylling (arve.kylling@nilu.no)

**Abstract.** Three-dimensional (3D) cloud structures may impact atmospheric trace gas products from ultraviolet-visible (UV-VIS) sounders. We used synthetic and observational data to identify and quantify possible cloud-related bias in $NO_2$ tropospheric vertical column densities (TVCD). The synthetic data were based on high-resolution large eddy simulations which were input to a 3D radiative transfer model. The simulated visible spectra for low-earth orbiting and geostationary geometries were analysed with standard retrieval methods and cloud correction schemes that are employed in operational $NO_2$ satellite products. For the observational data the $NO_2$ products from the TROPOspheric Monitoring Instrument (TROPOMI) were used while the Visible Infrared Imaging Radiometer Suite (VIIRS) provided high spatial resolution cloud and radiance data. $NO_2$-profile shape, cloud shadow fraction, cloud top height, cloud optical depth, solar zenith and viewing angles, were identified as the metrics being the most important in identifying 3D cloud impacts on $NO_2$ TVCD retrievals. For a solar zenith angle less than about $40°$ the synthetic data show that the $NO_2$ TVCD bias is typically below 10%. For larger solar zenith angles both synthetic and observational data often show $NO_2$ TVCD bias on the order of tens of %. Specifically, for clearly identified cloud shadow bands in the observational data, the $NO_2$ TVCD appears low-biased when the cloud shadow fraction $> 0.0$ compared to when the cloud shadow fraction is zero.

## 1 Introduction

Operational retrievals of tropospheric trace gases from space–borne spectrometers are based on the use of 1D radiative transfer models. To minimize cloud effects, generally only partially cloudy pixels are analysed using simplified cloud contamination treatments based on radiometric cloud fraction estimates (e.g. Grzegorski et al., 2006; Stammes et al., 2008) and photon path length corrections based on oxygen collision pair ($O_2$–$O_2$) (Acarreta et al., 2004; Veefkind et al., 2016) or $O_2$–A absorption band measurements (for example the FRESCO and OCRA/ROCINN algorithms, see Koelemeijer et al., 2001; Wang et al., 2008; Loyola et al., 2018; Liu et al., 2021). In reality however the impact of clouds can be much more complex, involving unresolved sub-pixel clouds, scattering of clouds in neighbouring pixels and cloud shadow effects. In a model study Merrelli

et al. (2015) showed that 3D radiation scattering from unresolved boundary layer clouds may give significant biases in Orbiting Carbon Observatory-2 (OCO-2) retrievals of $CO_2$ concentration. Massie et al. (2017, 2020) provided observational evidence of 3D cloud effects in OCO-2 $CO_2$ retrievals and found them consistent with 3D radiative transfer simulations. For airborne and ground-based remote sensing Schwaerzel et al. (2020, 2021) have shown the importance of acccounting for 3D radiative

transfer in air mass factor calculations when the atmosphere cannot be assumed to be horizontally homogeneous and when buildings are present.

In general, space-borne measurements of trace gases may be cloud contaminated and the presence of clouds may result in both positive and negative biases. It is thus vital to quantify the impact of clouds on trace gas retrievals and, if possible, envisage correction methods. To exclude cloud contaminated pixels from analysis is not a viable option as this, for example,

may give bias in long-term averages, as shown by Geddes et al. (2012) for $NO_2$. Furthermore, the loss in data coverage when excluding partially cloudy scenes may become critical, especially for low/medium resolution sensors and for regions where the probability of cloud occurence is high (e.g. Germany and other western European countries).

This paper is one of a series of three papers discussing the impact of 3D cloud structures on the atmospheric trace gas products from ultraviolet-visible (UV-VIS) sounders. The first paper by (Emde et al., 2021) describes the synthetic data which

is based on 3D radiative transfer model simulations utilizing realistic 3D clouds as input and is designed for validation of remote sensing trace gas retrievals. In this paper, the bias due to 3D clouds is quantified using both synthetic and observational data. Finally, in the third paper, Yu et al. (2021) discuss trace gas retrieval and mitigation strategies in the presence of 3D cloud structures.

We chose to study $NO_2$, as it is an important measure of air quality and a key tropospheric trace gas measured by the

atmospheric Sentinels (sentinel.esa.int). The synthetic data, which ignore stratospheric $NO_2$, were used to identify the cloud situations that give bias in tropospheric $NO_2$ retrievals. The observational satellite data from TROPOMI were used to investigate the presence of 3D cloud radiative transfer effects in real data. While focus is on tropospheric $NO_2$, the results are expected to be valid for other UV-VIS derived trace gas products.

From the HD(CP)2 (hdcp2.eu) project large eddy simulations (LES) based on the ICOsahedral Non–hydrostatic atmosphere

model (ICON Dipankar et al., 2015; Zängl et al., 2015) are available for a region including Germany, the Netherlands and parts of other surrounding countries. This unique synthetic data set provide realistic input for 3D radiative transfer modelling. Hence, we adopted the study region to be the area covered by the LES. This region also covers part of the footprints of the upcoming Sentinel-4 and Sentinel-5 missions. Using the LES output, satellite radiances were simulated. To include 3D radiative effects we used the 3D MYSTIC Monte Carlo radiative transfer model (Mayer, 2009; Emde et al., 2011) to generate

synthetic observations (Emde et al., 2021). From the simulated spectra the slant $NO_2$ column amounts were obtained using the QDOAS retrieval algorithm (Blond et al., 2007; De Smedt et al., 2008; Yu et al., 2021). The slant column densities were converted to vertical column densities using layer air mass factors (AMF). The AMFs included state-of-the-art independent pixel approximation cloud correction schemes (using the $O_2$-$O_2$ or $O_2$-A band) and were calculated with the VLIDORT 1D radiative transport model (Spurr, 2006). The retrieved $NO_2$ using standard 1D algorithms was compared to the input to the 3D

radiative transfer simulations and possible 3D radiative effects were identified and quantified.

The observational satellite data were used to investigate the presence of 3D cloud radiative transfer effects in real data. $NO_2$ tropospheric vertical column density data were taken from the TROPOMI/S5P operational L2 tropospheric column product. S5P does not include an imager for cloud information. However, it flies in tandem with the Suomi National Polar-orbiting Partnership (S-NPP) satellite. The S-NPP payload includes the Visible Infrared Imaging Radiometer Suite (VIIRS) instrument which may be used as an imager for TROPOMI. We utilized data from both sensors.

The paper first discusses the synthetic and observational data sets, section 2. This is followed by a description of the various metrics used to identify cloud impacts, section 3. In section 4 the results are presented and discussed before the paper ends by some concluding remarks and an outlook in section 5.

## 2   Data

The study area covers approximately Germany, the Netherlands and parts of other surrounding countries. For this region we have generated synthetic satellite data using high resolution LES data as input for 3D radiative transfer simulations.

### 2.1   Synthetic satellite data

The synthetic satellite data comprise results from three sources: 1) high-resolution LES cloud data; 2) 3D radiative transfer modelling of satellite radiances with the LES cloud data as input; and 3) $NO_2$ retrieval using the synthetic satellite radiances.

Within the HD(CP)2 project LES were made with the ICOsahedral Non–hydrostatic atmosphere model (ICON; Dipankar et al., 2015; Zängl et al., 2015) which simulated realistic liquid and ice clouds. The results have a spatial resolution of approximately $1.2 \times 1.2$ km$^2$ for a region including Germany, the Netherlands and parts of other surrounding countries and was validated against ground and satellite–based observational data by Heinze et al. (2017). Several weeks of simulations are available including all kinds of weather situations in Europe. We, however, utilize only one from 29 July 2014, 12:00 UTC, due to the computational burden of the radiative transfer simulations. The simulated scene includes all cloud types that are typical for Europe, such as shallow cumulus, cirrus, stratus, and also convective clouds. While focus is on Europe, the results/methods are expected to be general and thus applicable elsewhere.

The LES results were input to the 3D MYSTIC Monte Carlo radiative transfer model (Mayer, 2009; Emde et al., 2011) run within the libRadtran package (Mayer and Kylling, 2005; Emde et al., 2016) to generate synthetic observation spectra in the visible spectral range from 400–500 nm and in the $O_2$-A band region from 755–775 nm (for further details see: Emde et al., 2021; Yu et al., 2021). The spatial resolution of the simulated sensor was set to approximately 7×7 km, corresponding to 98×104 pixels for the full LES domain. Note that each simulated sensor pixel includes 36 cloud pixels, hence the simulations include sub-pixel cloud inhomogeneity. The synthetic spectra were input to a $NO_2$ retrieval algorithm which included two steps: first a differential optical absorption spectroscopy (DOAS) fit was performed using the QDOAS retrieval algorithm (Blond et al., 2007; De Smedt et al., 2008) to get the $NO_2$ slant column densities; second the slant column densities were converted to vertical column densities using layer air mass factors based on the VLIDORT 1D radiative transport model (Spurr, 2006). The fitting window was between 425 and 495 nm, similar to the one used by Richter et al. (2011). The air mass factor was calculated at

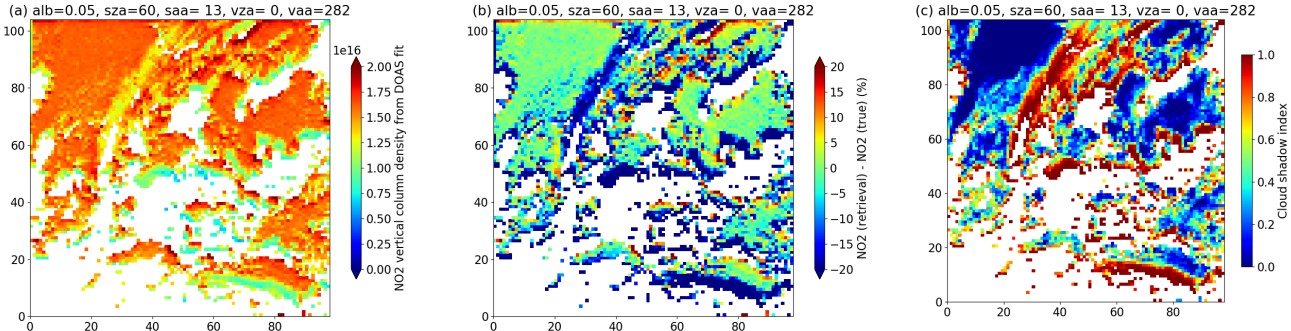

**Figure 1.** (a) The retrieved NO$_2$ TVCD for the low-earth orbit geometry case with albedo=0.05, solar zenith angle=60°, solar azimuth angle=13°, satellite viewing angle=0°, and satellite azimuth angle=282° (identical for all pixels). (b) The difference between the retrieved and the true NO$_2$ TVCD. (c) The cloud shadow index, see section 3.3 for details. The units on the axes are in pixel numbers. White pixels are cloudy regions for which the retrieval was not performed.

the middle of the fitting window, that is 460 nm. Cloud corrections were made using both O$_2$-O$_2$ and O$_2$-A band (FRESCO, OCRA/ROCINN) based methods (Yu et al., 2021). An example of retrieved NO$_2$ TVCD for a synthetic case is shown in Fig. 1a. Note that the "true" NO$_2$ is constant over the scene with column density of 1.6×10$^{16}$molec/cm$^2$ corresponding to an European tropospheric polluted NO$_2$ profile from Levelt et al. (2009). Thus any differences between the retrieved and "true"
5   NO$_2$ TVCDs are due to the presence of clouds, Fig. 1b. The NO$_2$ retrieval is further discussed by Yu et al. (2021).

   While we discuss our synthetic results in connection with observational results from satellites in low-earth-orbit (LEO), simulations were also made for a geostationary orbit (GEO). Azimuth and zenith viewing and solar angles were chosen to resemble geometries for the study region when viewed by the TROPOspheric Monitoring Instrument (TROPOMI, Veefkind et al., 2012) and the future Ultra-violet Visible Near-infrared (UVN, https://sentinel.esa.int/web/sentinel/missions/sentinel-4)
10   instrument to be in geostationary orbit. In total 15 and 36 combinations of viewing and solar angles were simulated for the GEO-case and LEO-case, respectively (Emde et al., 2021). The surface was assumed to be snow free and with constant albedo to simplify the interpretation of the results. In the visible (400-500 nm) simulations were made with albedos of 0, 0.05 and 0.2, while in the O$_2$-A band region an additional albedo of 0.5 was included to account for the potentially larger albedo in this part of the spectrum. Combining the sun-sensor geometries and visible albedo values our simulated data set thus include a total of
15   45 GEO-cases and 108 LEO-cases.

## 2.2   Observational satellite data

Both observational satellite spectrometer and imager data were utilized. NO$_2$ TVCD data were taken from the TROPOMI/S5P operational L2 NO$_2$ tropospheric column product. TROPOMI/S5P was launched 2017-10-13 and L2 NO$_2$ data are available from 2018-06-28. It passes the equator at 1330 local time. Imager data were taken from VIIRS and we used both L1b and

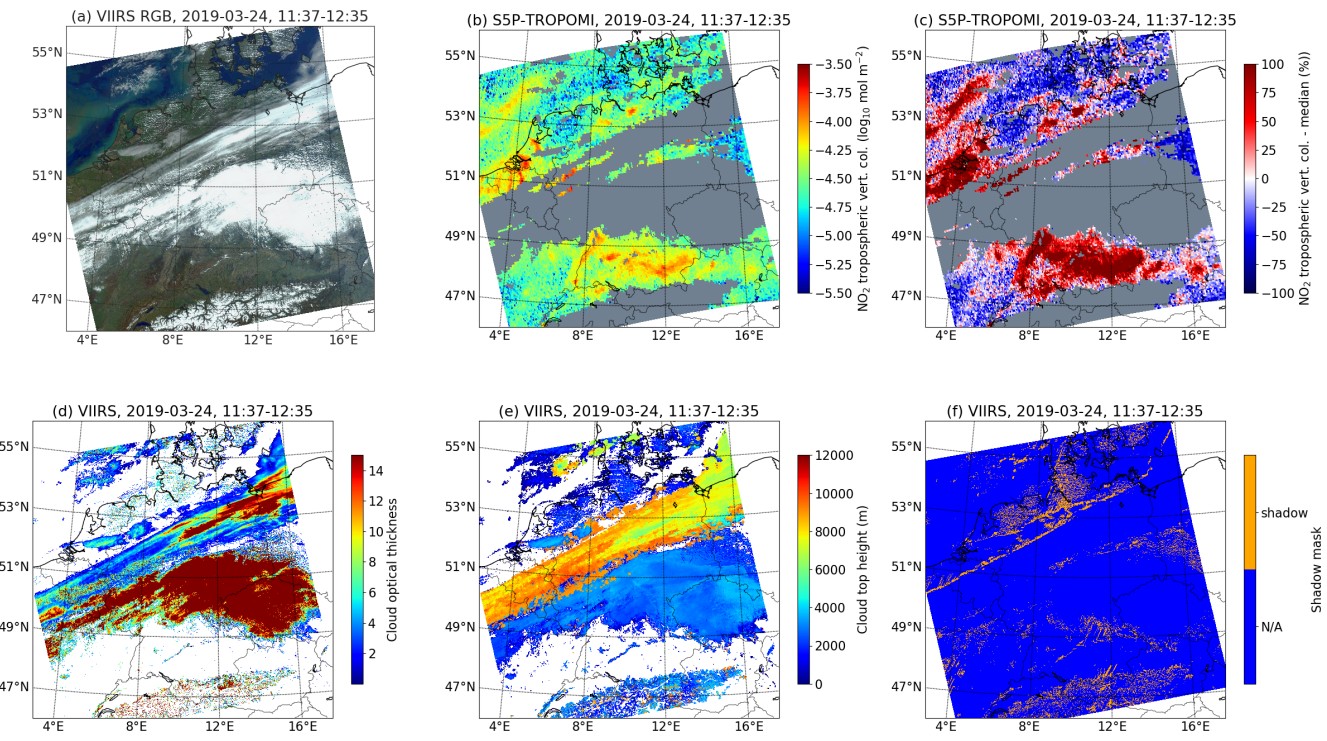

**Figure 2.** (a) RGB composite of VIIRS bands M3, M4 and M5 (centred at 0.488, 0.555 and 0.672 $\mu$m). (b) The tropospheric NO$_2$ vertical column density from TROPOMI. Only pixels with data quality value $> 0.95$ are displayed. (c) The percentage difference of the NO$_2$ column from the median over the study area. (d) The VIIRS cloud optical thickness. (e) The VIIRS cloud top height. (f) The VIIRS cloud shadow mask.

L2 data. The L1b reflectances were used for RGB plots and to produce various metrics, see section 3. The L2 data include various cloud products of which we used the cloud mask, cloud shadow, cloud optical thickness and cloud top height products. The TROPOMI data includes the latitude and longitude of the corners of each TROPOMI pixel. This information was used to identify VIIRS pixels within each TROPOMI pixel.

5      Data within the study region roughly covering Germany and parts of neighbouring countries were collected for the periods 28 June 2018 - 15 October 2018 and 1 March 2019 - 30 June 2019 when the sun was sufficiently high on the horizon to avoid problems with NO$_2$ retrievals for low sun (solar zenith angle $< 60°$). The NO$_2$ and cloud situations generally vary a lot for this region. In Fig. 2 an example of NO$_2$ data from TROPOMI, (Fig. 2b,c) and cloud optical depth (Fig. 2d), cloud top height (Fig. 2e) and cloud shadow fraction (Fig. 2f) from near simultaneous and VIIRS overpasses of Germany, is shown. A distinct

10 cloud shadow band is seen in Fig. 2a,d,e starting at about $50°$N, $3°$E and extending in the east/north-east direction. This cloud shadow band case is further discussed below.

## 3 Methods

Several metrics were calculated to quantify cloud features and their possible connection with $NO_2$-biases due to 3D cloud effects.

### 3.1 Cloud geometric and radiance fractions

Cloud fractions may be defined in several ways. We calculate the geometric cloud fraction, $CF_g$, the radiometric cloud fraction, $CF_r$, and the weighted radiometric cloud fraction, $CF_w$.

The geometric cloud fraction for a TROPOMI pixel is defined as

$$CF_g = \sum CM_i/N, \tag{1}$$

where the sum is over all $N$ VIIRS pixels within the TROPOMI pixel and $CM_i$ is the VIIRS cloud mask where $CM_i = 1$ for pixels identified as cloudy and $CM_i = 0$ otherwise.

The radiometric cloud fraction is the fraction of measured radiance reflected from clouds in a pixel (see also Grzegorski et al., 2006)

$$CF_r = \begin{cases} 0 & R \le R_s \\ \frac{R - R_s}{R_c - R_s} & R_s < R < R_c \\ 1 & R \ge R_c \end{cases} \tag{2}$$

Here $R$ is the observed reflectance, $R_s$ is the reflectance for a cloudless sky and $R_c$ the reflectance for an opaque cloud. For the $O_2$-$O_2$ cloud correction, $CF_r$ is calculated based on the reflectance at 460nm which is in the middle of the DOAS fitting window for $NO_2$. For the FRESCO algorithm, $CF_r$ is determined by the reflectance in the 758-759nm window band. Further details are described by Yu et al. (2021). We also define an average radiometric cloud fraction $CF_r^{VIIRS}$ using the average of $CF_r$ calculated for each VIIRS M3 band pixel, centred at 0.488 $\mu$, within a TROPOMI pixel:

$$CF_r^{VIIRS} = \sum CF_{r,i}/N, \tag{3}$$

where the sum is over all $N$ VIIRS pixels within the TROPOMI pixel.

Finally, the weighted radiometric cloud fraction is defined as (Yu et al., 2021)

$$CF_w = \frac{CF_r R_c}{CF_r R_c + (1 - CF_r) R_s} \tag{4}$$

### 3.2 Cloud mask

No cloud mask was available for the LES based synthetic data. In principle this may be calculated from the cloud optical thickness assuming some treshold. However, we chose to calculate a cloud mask similar to how it is done for several operational satellite products. We thus assume that pixels with reflectance larger than some threshold, $R > R_t$, are cloudy. The cloud mask

was calculated from high spatial resolution radiance simulations at 0.55 $\mu$m of the synthetic cases described by Emde et al. (2021). A threshold of $R_t = 0.25$, similar to Heinze et al. (2017), was adopted. It is noted that Barker et al. (2017) used $R_t = 0.15$ for the GOES-13 0.65 $\mu$m band. Yang and Di Girolamo (2008) has shown that there may be overlap between clear and cloudy pixels and hence some mis-classification is unavoidable. Our slightly higher value of $R_t$ potentially includes more cloud contaminated pixels, but also reflects that we are mainly working over land where surface albedo is higher than over ocean.

## 3.3 Cloud shadow fraction and cloud shadow index

The VIIRS cloud shadow mask algorithm is geometry-based and described by Hutchison et al. (2009). They compared the MODIS MOD35 product which uses spectral signatures to identify cloud shadows with geometry-based approaches and states that the latter "are far superior to those predicted with the spectral procedures". A cloud shadow detection algorithm using TROPOMI data only have been described by Trees et al. (2021). It was, however, not available for this study.

The cloud shadow fraction, $CSF$, for a TROPOMI pixel is defined as

$$CSF = \sum CSM_i / N, \tag{5}$$

where the sum is over all $N$ VIIRS pixels within the TROPOMI pixel and $CSM_i$ is the VIIRS cloud shadow mask where $CSM_i = 1$ for pixels identified as cloud shadow and $CSM_i = 0$ otherwise.

For the LES we define the cloud shadow index ($CSI$) as a surrogate for a cloud shadow product as follows:

$$CSI = E_0 - E_{\theta,\phi} / \cos(\theta) \tag{6}$$

Here $E_0 = 1$ is the direct transmittance at the surface for an atmosphere with no molecular absorption nor clouds or aerosol. The direct transmittance, $E_{\theta,\phi}$, at the surface for a solar zenith angle of $\theta$ and solar azimuth angle $\phi$, was calculcated with the MYSTIC model. It also does not include molecular absorption nor aerosol, however, it includes clouds, which are fully accounted for in 3D geometry and with the 3D cloud fields in higher spatial resolution than the TROPOMI pixel size. The $CSI$ is zero for pixels with no cloud shadow and 1 if a pixel is fully in the cloud shadow. The $CSI$ clearly and unambigously identify cloud shadow regions as it is based solely on sun, cloud and surface geometry. An example of the $CSI$ is given in Fig. 1c.

## 3.4 H-metric

For a TROPOMI pixel the H-metric is the standard deviation of VIIRS band reflectances divided by the mean of the VIIRS reflectances within the TROPOMI pixel. It is an estimate of the variation of radiance within the TROPOMI pixel and has been used earlier by for example Massie et al. (2017) as metric for the impact of 3D clouds on $CO_2$ retrievals. In the presence of sub-pixel clouds the H-metric is expected to increase as the horizontal cloud inhomogeneity increase. In the absence of clouds the H-metric is expected to be small. However, for cloud free pixels with large variations in surface albedo the H-metric may still be large, and conversely, for a completely cloudy pixel the H-metric may be small. Thus, it may not be used without care to

unambiguously identify cloud inhomogeneity. From the VIIRS data we calculate the H-metric for the M3, M4 and M5 bands centred at 0.488, 0.555 and 0.672 $\mu$m respectively.

## 3.5 Absorbing aerosol index

Clouds have an effect on the UV absorbing aerosol index (AAI). It is thus of interest to investigate possible relationships between the AAI and the cloud shadow fraction and the $NO_2$ TVCD. The AAI is a measure of the UV color of a cloud-, aerosol- and shadow-free 1-D atmosphere-surface model with respect to the measured UV color (de Graaf et al., 2005). When absorbing aerosols are present, the AAI tends to be positive, while the AAI is approximately zero or negative in the presence of clouds (see e.g. Kooreman et al., 2020; Penning de Vries et al., 2009a). We use the TROPOMI AAI product to investigate relationships between the AAI and the $NO_2$ TVCD.

## 4 Results

For both the synthetic and observational data the aim is to compare the $NO_2$ TVCD for cloud affected cases with the true $NO_2$ TVCD and identify potential biases. This is straightforward for the synthetic data as the true $NO_2$ TVCD is known. For the observational data the true $NO_2$ TVCD is in general not known. One approach to estimate the true $NO_2$ TVCD from the observational data is discussed in section 4.2, which also include the analysis of the observational data.

### 4.1 Synthetic satellite data

For the synthetic data the fully cloudy pixels were not simulated due to the computational burden, see Emde et al. (2021) for details. To further filter for the presence of clouds the $NO_2$ TVCDs were retrieved for pixels where the radiometric cloud fraction $CF_r < 0.3$ for all LEO and GEO cases. The retrieval tries to correct for the presence of remaining clouds by including a standard photon path length correction based on absorption by the oxygen collision pair $O_2$-$O_2$ or the $O_2$-A band as described by Yu et al. (2021). An example of the retrieved $NO_2$ TVCD and the bias is shown in Figs. 1a and b, while the cloud shadow index is shown in Fig. 1c.

For three LEO cases the difference between the retrieved and "true" $NO_2$ TVCDs versus the cloud shadow index is shown in Fig. 3. The cloud shadow impact is seen to increase as the solar zenith angle increases. The number of pixels with $NO_2$ TVCD differences $< -20\%$ is 0.1% for a solar zenith angle of $20°$ (Fig. 3a), 4.% for $40°$ (Fig. 3b) and 20.3% for $60°$ (Fig. 3c). As the solar zenith angle increases a linear relationship appears between the $NO_2$ TVCD difference and the $CSI$ as indicated by the black lines and corresponding $R^2$ values. The increase of the cloud shadow impact with increasing solar zenith angle is due to the increase in cloud shadow size for larger solar zenith angles from geometrical reasons, see Emde et al. (2021) for a detailed discussion.

For the cloud correction of the AMF the weighted radiometric cloud fraction, $CF_w$, is used as described by Yu et al. (2021). The effect of increasing $CF_w$ is shown in Fig. 4 where the $NO_2$ AMF bias (1D AMF - 3D AMF) is plotted as a function of $CF_w$ for all GEO and LEO geometries. The retrieval bias is binned in $CF_w$ bins with 2% steps which gives more than 5000

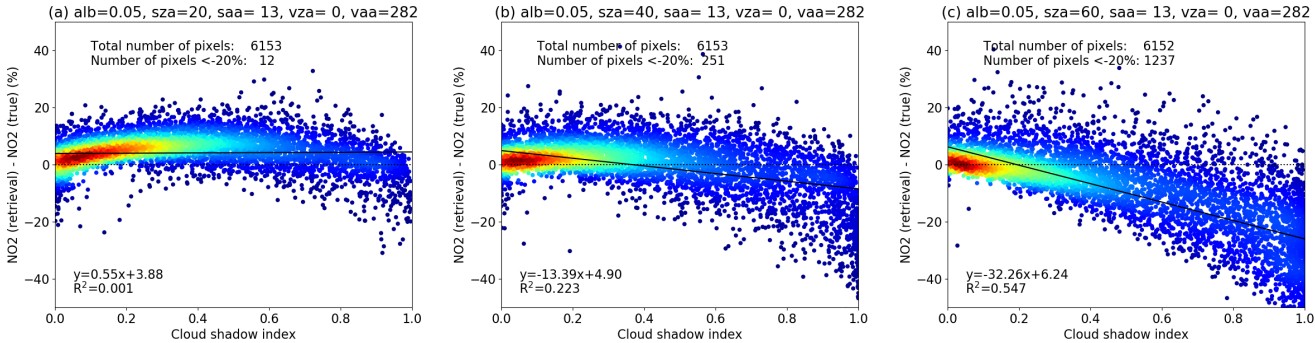

**Figure 3.** The difference between the retrieved and "true" $NO_2$ TVCDs versus the cloud shadow index for three low earth-orbiting-geometry cases. Results are shown for cloud filtered pixels and solar zenith angles of (a) 20, (b) 40 and (c) 60°, albedo=0.05, solar azimuth angle=13°, satellite azimuth angle=0°, and satellite viewing angle=282°. The black lines are linear fits and the fit parameters and $R^2$ are given in the individual plots. The data in (c) are for the case presented in Fig. 1.

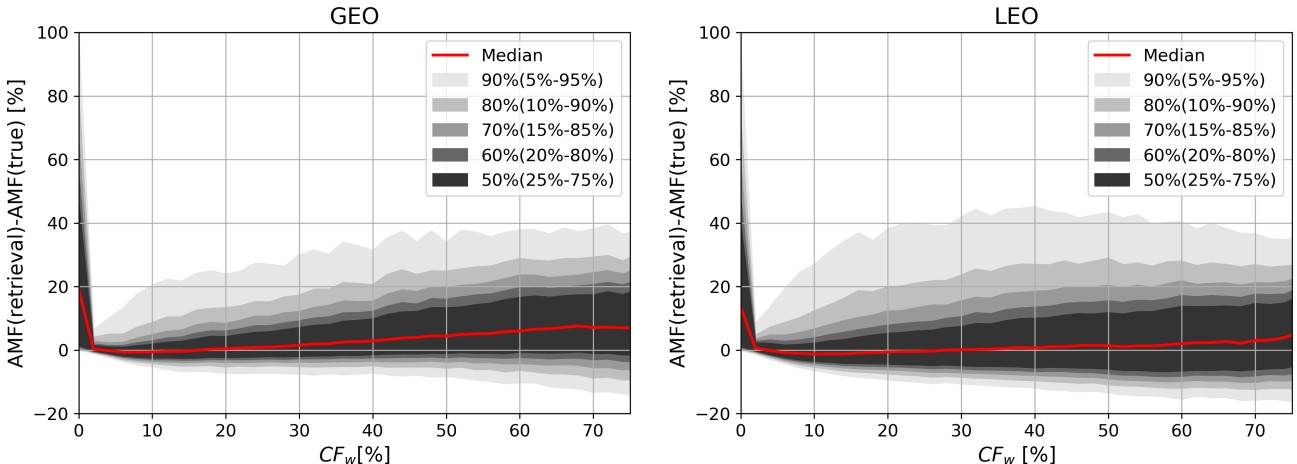

**Figure 4.** Distribution of the $NO_2$ AMF bias as a function of the weighted radiometric cloud fraction, $CF_w$, for geostationary and low earth orbiting geometries. Results are shown for the retrieval using the $O_2$-$O_2$ cloud correction. Results are similar for the $O_2$-A band based cloud correction.

pixels in each bin when the $CF_w < 75\%$. The bias may be due to several causes. The $NO_2$ profile and surface albedo input gives differences between the LIDORT, used in the retrieval, and MYSTIC RTM, used for the 3D simulations, on the order of 1% (Emde et al., 2021; Yu et al., 2021). As discussed by Yu et al. (2021), the $NO_2$ retrieval error due to the use of a simple cloud correction scheme is generally less than 20% for nearly cloud-free cases ($CF_w < 50\%$). Therefore, pixels with a $NO_2$ retrieval bias >20% are likely to be affected by cloud shadow effects. For both geometries the $NO_2$ AMF bias is high (median of 20%) and with a large range (0-65%) for $CF_w < 1\%$. For these clear pixels there is a significant number of pixels with the

retrieval bias > 20% and this is attributed to cloud shadow effects. The bias decreases to 0% when the $CF_w$ is between 1-3%. The bias range increases slightly with inreasing $CF_w$: about 75% of the pixels have a positive bias for large $CF_w$ for GEO geometry, while there are comparatively more pixels with a negative bias for LEO geometry. The solar zenith angle is the same for the LEO and GEO cases. However, the solar azimuth angle and the sensor viewing zenith and azimuth angles are different.

For the LEO and GEO geometries studied, see Emde et al. (2021) for details, the sun is to the south of the study region. This implies that a relatively large portion of cloud shadows are on the northern sides of the clouds. These cloud shadows are partly hidden from GEO satellites but may be visible from LEO satellite instrument with a nadir view of Earth, thus giving different sensitivity to cloud shadows for LEO and GEO geometries. More than 15% of the pixels have a bias larger than 20% for $CF_w > 50\%$ for both GEO and LEO geometries.

Figure 5 shows the number of pixels that satisfy these conditions for all the LEO and GEO geometries. For the geostationary geometry, the number of pixels with retrieval bias > 20% is up to about 1000 or 10.6% (out of 9400 pixels for a single case). This number increases with surface albedo and SZA, the difference between surface albedos of 0.05 and 0.2 is about 150 to 400 pixels. As discussed above the increase with solar zenith angle is due to larger cloud shadow due to geometrical reasons. With respect to the albedo dependence, Emde et al. (2021) showed that the relative 3D-1D difference in reflectance increase with

increasing surface albedo. Furthermore, Yu et al. (2021) showed that this gives increased differences in $O_2$–$O_2$ and FRESCO cloud-corrected AMFs with increasing surface albedo. This number also strongly depends on the solar azimuth angle (SAA), and the difference between SAAs of 315° and 45° is more than 25%. For low earth orbiting geometry, the number of pixels with bias > 20% increases with surface albedo and SZA as well, and it is up to 1600 (17%) for high surface albedo of 0.2 and SZA of 60°.

To identify the localisation of the pixels with $NO_2$ AMF bias > 20%, maps of the number of cases with $NO_2$ AMF bias > 20% were made as shown in Fig. 6. Generally, the majority of pixels with large $NO_2$ AMF bias are found in cloud free regions close to cloud edges. For a solar zenith angle around 40° most pixels will have an AMF bias below 20% if the distance to the cloud edge is more than about 10 km. For a solar zenith angle of 60° this distance increase to about 20 km. The distance depends on a number of factors such as cloud top height, cloud optical depth and surface albedo. This is further discussed and

quantified for box clouds in the accompanying paper by Yu et al. (2021). Figure 7 shows maps of the maximum retrieval bias for each pixel. As above, cloudy data ($CF_w > 50\%$) are excluded from the analysis. The maximum bias is usually less than 60% over the northwest region (x=30-50, y=70-100) where there is an ice cloud with small optical thickness ($< 10$). In the center of the map, the bias is often higher than 100%, the pixels around are covered by a convective cloud with large vertical extent and optical thickness $> 100$. Fig. 8 shows under which geometry the maximum bias is observed. Pixels with maximum

biases < 20% are not shown. In general, the maximum bias is obtained at high SZA (60°), and for GEO cases, the bias also depends on the solar azimuth angle. Maximum biases are found on the east/west side of the cloud when the sun is in the west (SAA=-90°)/east (SAA=90°). Note that this dependency for GEO geometry is particular for the region, and thus solar and viewing conditions, studied. This dependency is not seen for the LEO geometry as for LEO geometry the local daily revisit time is the same and thus also the solar azimuth.

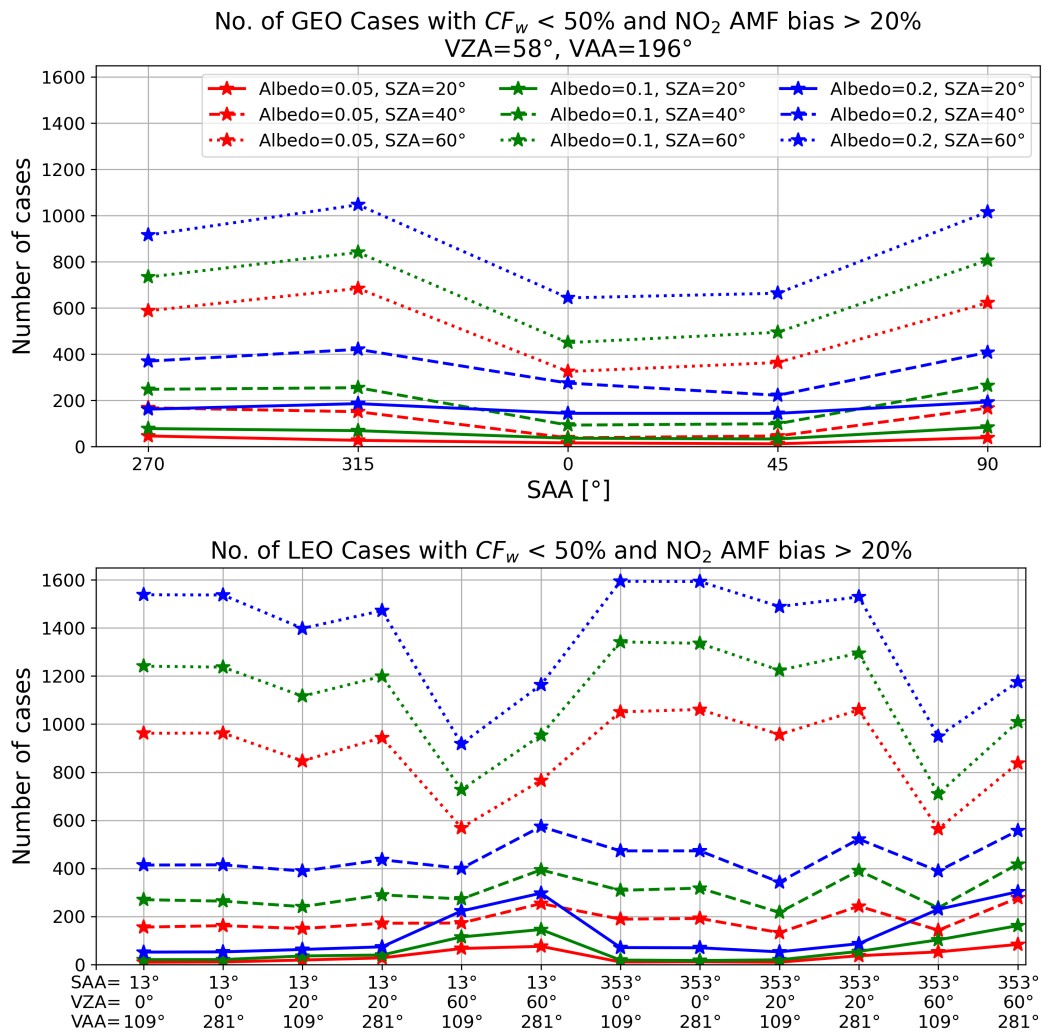

**Figure 5.** Number of pixels with weighted radiometric cloud fraction $CF_w < 50\%$ and NO$_2$ AMF bias > 20% for geostationary (top) and low earth orbiting (bottom) geometries. The data points are connected by lines for increased readability. Different line styles indicate different SZA while line color are used for the surface albedo as given in the annotation. For the geostationary geometry the solar azimuth angle (SAA) for each case is given on the x-axis label, while for the low earth orbiting geometry the viewing zenith angle (VZA) and viewing azimuth angle (VAA) is given in addition. The total number of pixels for a single case is 9400.

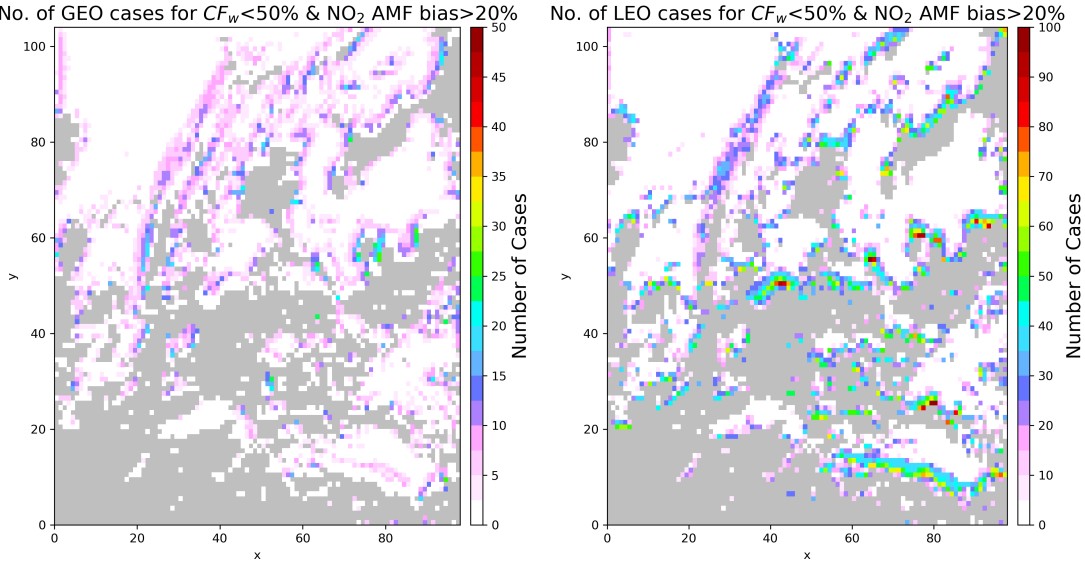

**Figure 6.** The number of cases with $CF_w < 50\%$ and $NO_2$ AMF bias $> 20\%$ for each pixel for GEO (left) and LEO geometries (right). The x- and y-axes represent pixel number in the longitude/latitude direction. Grey shaded pixels indicate cloudy pixels.

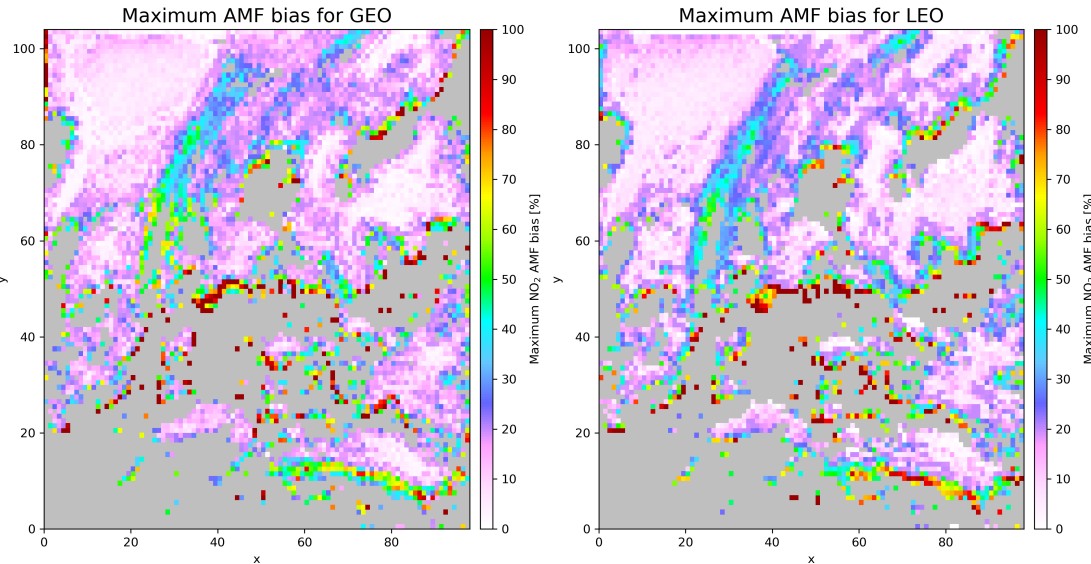

**Figure 7.** Maximum $NO_2$ AMF bias for each pixel (left: geostationary geometry; right: low earth orbiting geometry).

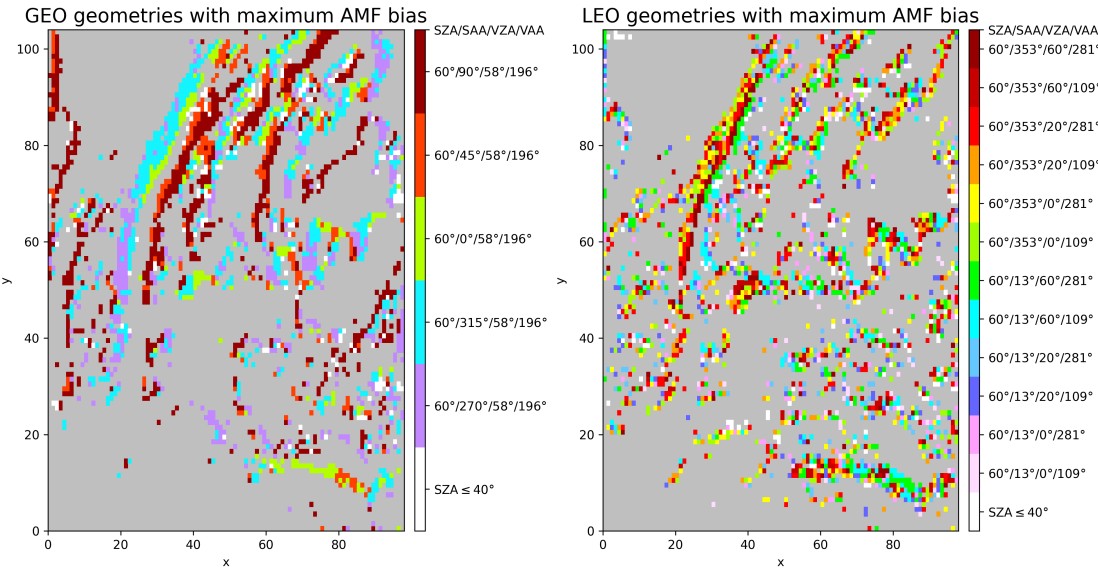

**Figure 8.** Geometry for the largest AMF bias for each location (left: geostationary geometry; right: low earth orbiting geometry). Only the pixels with maximum bias $> 20\%$ are shown.

To summarize the overall differences between the retrieved and true NO$_2$ TVCDs for all LEO and GEO cases, the number $n$ of pixels with differences $d$: $d < -20; -20 < d < 20; -10 < d < -10; d > 20\%$ were calculated. These results are summarized in Fig. 9 for geostationary and low-earth-orbiting geometries. For the geostationary geometry, between 87.6$\pm$0.6% (solar zenith angle, SZA=20°) and 64$\pm$2.5% (SZA=60°) of the retrieved NO$_2$ TVCD are within $\pm$10% of the "true" column for an albedo of 0.0 (black solid lines, Fig. 9a). These numbers decrease by about 8-10% points for larger surface albedos of 0.05 to 0.2 (dotted and dashed lines). The number of pixels within $\pm$20% decrease from 98.8$\pm$0.4 to 79.4$\pm$2.4 as the SZA increase from 20° to 60° for an albedo of 0.0 (green solid lines, Fig. 9a). The number of pixels with differences $< -20\%$ increase from about 1.0% for low albedo and SZA=20° to up to 17% for high albedo and SZA=60° (red lines Fig. 9a). The number of pixels with differences $> 20\%$ is zero for SZA=20° and never above 3.3% for other SZA. The overall bias is bias -0.9%. The variation with solar azimuth angle is small for SZA=20° and SZA=40°, but increase for SZA=60°. The viewing angles are constant for the geostationary cases and thus no variations are seen.

For the low-earth-orbiting geometry between 82.6$\pm$3.3% (SZA=20°) and 58.9$\pm$2.1% (SZA=60°) of the retrieved NO$_2$ TVCDs are within $\pm$10% of the true column for an albedo of 0.0 (black solid lines, Fig. 9b). These numbers decrease by about 7% points for larger surface albedos. The number of pixels with differences within $\pm$20% decrease from 97.0$\pm$3.1 to 76.3$\pm$3.5% as the SZA increases from 20° to 60° for an albedo of 0.0 (black solid lines, Fig. 9b). The number of pixels with differences $< -20\%$ increase from about 2.6$\pm$2.7% for low albedo (0.0) and high sun (SZA=20°) to up to 22.4$\pm$2.4% for high albedo (0.2) and low sun (SZA=60°, solid and dashed red lines, Fig. 9b). The number of differences $> 20\%$ are below 3.9% for all cases (blue lines, Fig. 9b). The average median bias is -0.5%. The differences shows no strong dependence on the

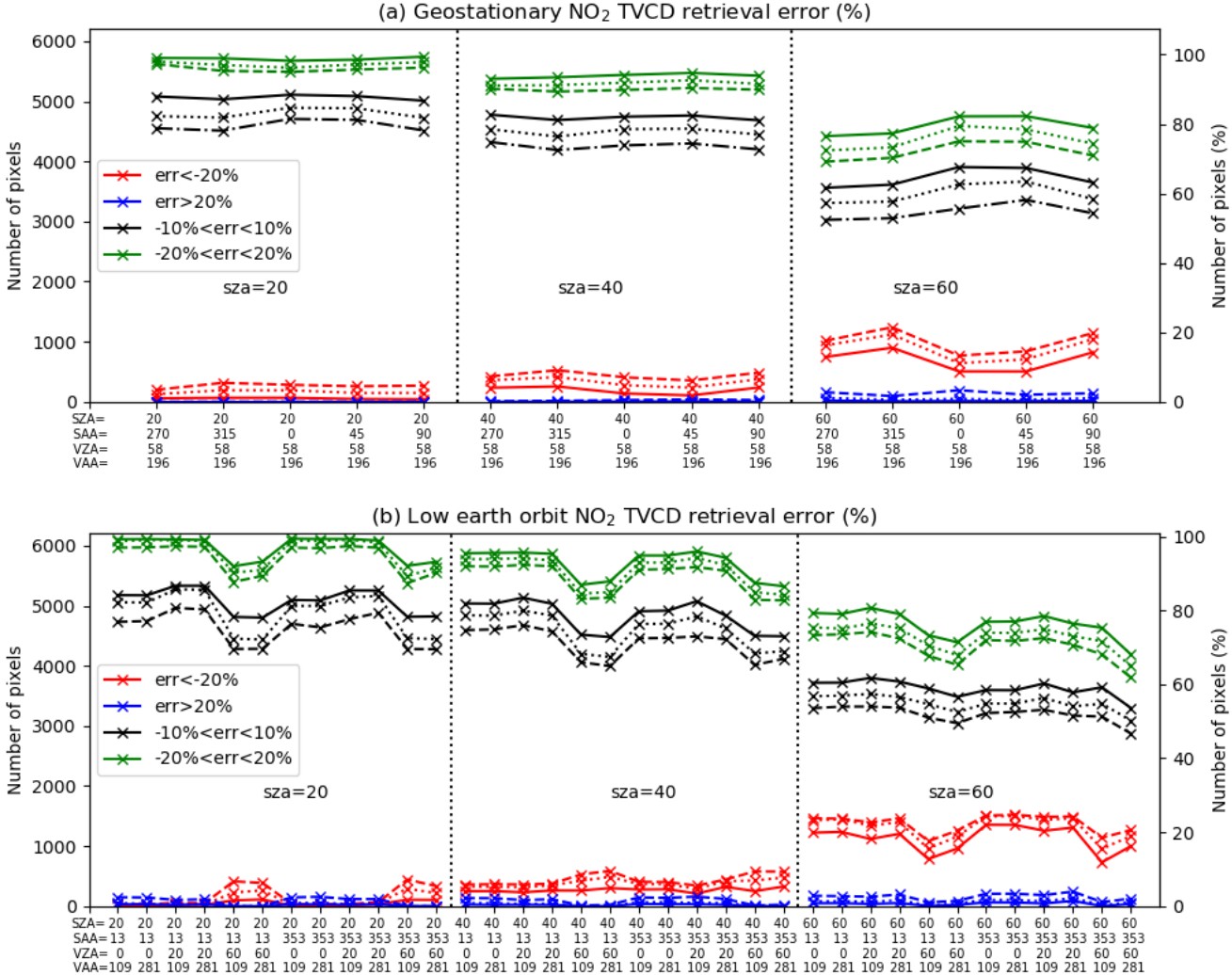

**Figure 9.** The number of pixels with NO$_2$ TVCD differences larger than 20% (blue lines) and less then -20% (red lines). The black and green lines are the nuber of pixels for which the differences are within ±10% and ±20%, respectively. Results are shown for geostationary (a) and low-earth-orbiting (b) geometries. The data points are connected by lines for increased readability, with solid lines being for an albedo of 0.0, dotted lines for 0.05, and dashed for an albedo of 0.2. The solar zenith (SZA) and azimuth angles (SAA) and the viewing zenith (VZA) and azimuth angles (VAA) are given on the x-axes. Note that the % is with respect to the total number of pixels for which a NO$_2$ retrieval has been done. This number is smaller than the number of pixels in the scene as cloud contaminated pixels are excluded from the retrieval.

viewing azimuth angle. However, the number of points within ±10% decrease by 4-8% for large viewing angles (VZA=60°) and SZA<= 40° (black lines, Fig. 9b).

These results show that the solar zenith angle is of prime importance for 3D cloud impacts and that the impact increases with increasing solar zenith angle. This is due to geometry reasons which cause the cloud shadow to increase as the solar zenith angle increases. Also, as the viewing zenith angle increases a larger, potentially cloud shadow impacted, horizontal surface area will be viewed due to geometry reasons and thus the cloud shadow effect increase with increased viewing zenith angle. Both under- and over-estimates of the $NO_2$ TVCD occur in pixels close to clouds. The underestimates are due to cloud shadows, thus the cloud shadow fraction is a cloud feature metric that may be used to identify affected pixels, Fig. 3. However, while for large solar zenith angles pixels affected by cloud shadows are mostly underestimated, overestimates occurr for all solar zenith angles and is mostly present for low cloud shadow fractions (Fig. 3) and increase for large surface albedo (blue dashed lines Fig. 9). Thus, cloud features in neighbouring pixels, such as cloud top altitude and cloud optical thickness, are also of importance (Emde et al., 2021).

According to box cloud simulations presented by Emde et al. (2021) and Yu et al. (2021), the cloud enhancement effect is of similar magnitude as the cloud shadow effect. For the synthetic data the enhancement effect is present, but is smaller in magnitude than the cloud shadow effect, see blue lines Fig. 9. This is most likely due to cloud enhancement affected pixels being identified as cloudy and thus not analysed. Also the synthetic data indicate that the smaller the solar zenith angle is, the larger are the chances for cloud enhancements between 5-10% for low earth orbit satellites (data not shown). The differences are largest for large satellite viewing angles (60°) and may thus indicate enhancement due to the satellite partly viewing sun-illuminated cloud sides. For geostationary geometry this effect is not present. This is due to differences in the solar azimuth angles for the two geometries. However, in magnitude, this effect has a much smaller impact than cloud shadows.

For large SZA (60°) and low albedo the number of pixels with $NO_2$ TVCD underestimated by more than $<-20\%$ is 12.0 and 18.3% for GEO and LEO geometries, respectively. For comparison Lorente et al. (2017) estimated the structural uncertainty (differences in retrieval methodology) in the tropospheric $NO_2$ AMF to be 42% over polluted regions and 31% over unpolluted regions. These differences are mostly driven by the uncertainty in the a-priori $NO_2$ profile, cloud properties and surface albedo. Thus, while smaller than the structural uncertainty, 3D cloud impacts constitute an additional significant error source for polluted conditions and large SZA. Note that the different cloud correction schemes ($O_2$-$O_2$ and $O_2$-A band) are normally within 10% (not shown) and this number may be interpreted as the level of uncertainty of the uncertainty introduced by these correction schemes.

## 4.2 Observational satellite data

### 4.2.1 Cloud shadow band cases

Based on the findings from the analysis of the synthetic data we searched TROPOMI and VIIRS data for cases with cloud shadows and large solar zenith angles. An example of a cloud shadow band is shown in Fig. 10 (pointed to by the red arrow in Fig. 10a). The $NO_2$ TVCD, Fig. 10b, is markedly higher in the Amsterdam area, but is otherwise overall slowly varying over the region. The H-metric, Fig. 10c, is sensitive to inhomogeneous clouds, but also to variations in the surface albedo, compare Fig. 10a and 10c. The geometric cloud fraction, based on the VIIRS cloud mask, shows larger variability than the average

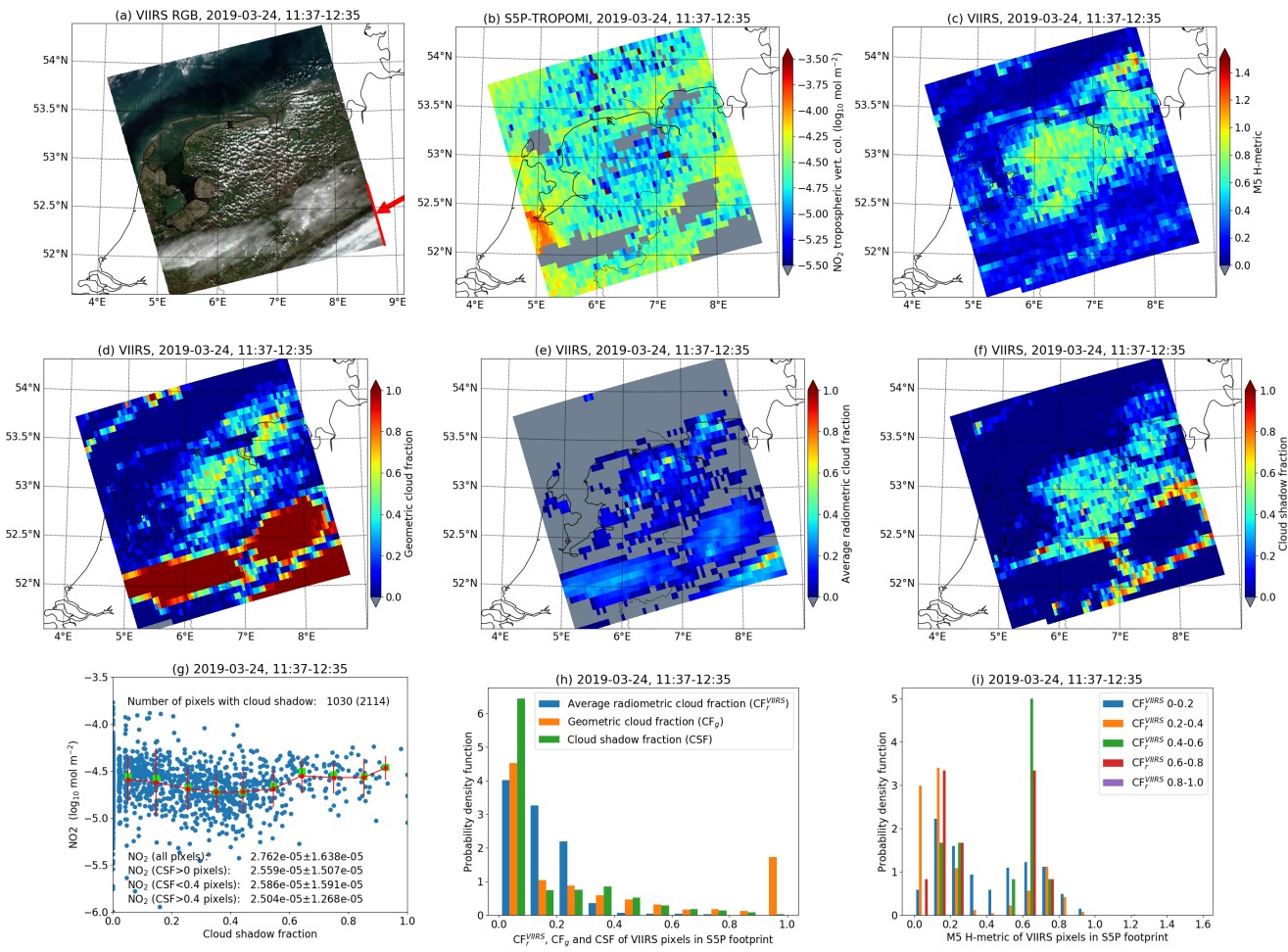

**Figure 10.** Examples of VIIRS and TROPOMI data and various CFMs for the cloud shadow band in Fig. 2. (a) VIIRS RGB. The red marked pixels correspond to the data shown in Fig. 11. (b) $NO_2$ TVCD from TROPOMI. (c) The H-metric from VIIRS band M5. (d) The cloud geometric fraction from VIIRS. (e) The cloud radiance fraction from VIIRS. (f) The cloud shadow fraction from VIIRS. (g) The $NO_2$ TVCD versus the cloud shadow fraction. The red lines is the average $NO_2$ TVCD with standard deviation (vertical lines). Lime green squares indicate the median $NO_2$ TVCD. (h) The distribution of the cloud radiance, cloud geometric and cloud shadow fractions. (i) The distribution of the H-metric for various cloud radiance fractions.

radiometric cloud fraction $CF_r^{VIIRS}$, as expected, compare Fig. 10d and 10e and see also the distributions of these quantities in Fig. 10h. The cloud shadow fraction, Fig. 10f, is between 0.2 and 0.5 in the area with scattered clouds. The cloud shadow fraction also clearly delineates the large cloud structures. The cloud shadow band has a width about the extent of 1-2 TROPOMI pixels. As the cloud shadow band and the TROPOMI pixels are not aligned this implies that the cloud shadow band at some

5 locations will be completely covered by one TROPOMI pixel and at other locations partly covered by two TROPOMI pixels.

This causes the oscillatory pattern seen in the geometric cloud fraction (Fig. 10d) and the cloud shadow fraction (Fig. 10f) in the cloud shadow band. The $NO_2$ TVCD versus the cloud shadow fraction is shown in Fig. 10g for all cloud shadow pixels. For a cloud shadow fraction $< 0.5$ the $NO_2$ TVCD decrease with increasing cloud shadow fraction. However, the decrease is well within the variability of the $NO_2$ TVCD (the red line with error bars indicates the average $NO_2$ TVCD $\pm$standard

deviation) and the scatter is too large to draw any conclusions about the dependence of the $NO_2$ TVCD on the cloud shadow fraction. The distribution of the H-metric for various $CF_r^{VIIRS}$ is shown in Fig. 10i. For $CF_r^{VIIRS}$ between 0.4 and 0.6 (green bars) maxima are found both for low (scattered clouds) and high (homogeneous clouds or surface) H-metrics indicating that the $CF_r^{VIIRS}$ may not unambiguously be used to identify scattered cloud cases. For the absorbing aerosol index no obvious dependence between the AAI and the $NO_2$ TVCD is present for this region, see Appendix A for further details.

For the cloud shadow band pointed to by the red arrow in Fig. 10a, further analysis is provided in Fig. 11. A RGB zoom in of the cloud shadow band is shown in Fig. 11b where red marks indicate pixels with cloud shadow. The VIIRS cloud top height and the TROPOMI $NO_2$ TVCD are shown in Figs. 11c and 11d. The cloud top height, cloud optical thickness and the cloud fraction is shown for row 267 in Fig. 11e. The cloud shadow is just north of a cloud band with optical thickness up to 10 (slant optical thickness of about 15) and an altitude between 9-10 km. North of the cloud shadow band there are also

some clouds at the same height but with a smaller optical thickness of around 1.5-2. These thinner clouds are not easily seen in the VIIRS RGB, Fig. 10a, but are present in the VIIRS cloud mask and thus the geometric cloud and averaged radiometric cloud fractions, Figs. 10d-e. The $NO_2$ TVCD for row 267 using FRESCO and OCRA/ROCINN cloud correction algorithms, is shown in Fig. 11f. The $NO_2$ TVCD inside the cloud shadow is low by about a factor $2-3$ compared with the $NO_2$ TVCD north of the cloud shadow. Fig. 11g displays the $NO_2$ TVCD south of the shadow, in the shadow and north of the shadow for

rows 262-269 and the average of these. A shadow pixel is defined as having a CSF $> 50\%$ and $CF_w < 50\%$. For row 263 no pixels satisfied this criteria and therefore no data is shown in the shadow region for this row. Pixels to be included in the regions north and south of the shadow band (up to 4 TROPOMI pixels north and south of shadow band), where required to have CSF $< 25\%$ and $CF_w < 50\%$. Except for rows 262 and 269, the $NO_2$ TVCD is smaller in the cloud shadow band compared to the $NO_2$ TVCD north of the cloud shadow. The $NO_2$ spatial varibility is large (Fig. 12.d), despite this, for the cloud shadow band

covered by rows 262-269, the $NO_2$ TVCD is on average reduced by 17%. There is no clear dependence of the $NO_2$ difference on the cloud shadow fraction. The cloud optical thickness of the cloud causing the cloud shadow varies little, the average cloud optical thickness being 9.0$\pm$1.6. Thus it is not possible to say anything about the dependence of the $NO_2$ difference on the cloud optical thickness for this case.

Another example of a cloud shadow band and 3D cloud effects on $NO_2$ retrieval is shown in Fig. 12. The VIIRS RGB image

(Fig. 12a) shows the cloud coverage over Northern Germany on December 30, 2019, when the solar zenith angle during the VIIRS and TROPOMI overpass time is larger than $70°$. The cloud shadow is just north of a large cloud band, and most of the northern region is completely cloudless, which is similar to the idealized box cloud cases presented by Emde et al. (2021) and Yu et al. (2021). The cloud height from west to east is 1 km to 8 km (Fig. 12c). The cloud optical thickness is not available for this case, but from the RGB the cloud is clearly optically thick enough to make the ground not visible from space. To look

for cloud shadow effects, we select the region within the red box in Fig. 12a. In Fig. 12e is shown the VIIRS reflectance and

cloud top height for TROPOMI row 394 as a function of latitude. The pixels are identified as cloudy, cloud shadow and clear based on the reflectance; with the reflectance being about 0.25 over the clear region, down to 0.18-0.24 in the cloud shadow, and higher than 0.25 for cloudy pixels. There are four TROPOMI pixels in the cloud shadow where the $NO_2$ TVCD is low by 20-60% compared with the $NO_2$ TVCD to the north and south of the cloud shadow (cloud and clear pixels). There is a slight difference in the $NO_2$ retrieval using different cloud corrections, Fig. 12f. In order to reduce the influence of the signal to noise ratio, the $NO_2$ TVCD is averaged over cloudless (four pixels near the cloud shadow), cloud shadow and cloudy pixels (four pixel near the cloud shadow) for rows 393 to 398, as shown in Fig. 12g. For the cloudy pixels, the difference is within 10% between $NO_2$ averaged over all the pixels and $NO_2$ averaged for the pixels with high quality retrieval ($CF_w < 50\%$). With the exception of the cloudy pixels south of the cloud band for row 396, all other cases show that the $NO_2$ TVCD in the cloud shadow is lower by 8-46% (average of 25%) compared with the $NO_2$ TVCD around the shadow. Here, the $NO_2$ TVCD around the shadow represents $NO_2$ retrieval unaffected by 3D cloud, which is an average of $NO_2$ over cloudy and cloudless pixels, in order to reduce the impact of spatial variation of $NO_2$.

The time difference between the VIIRS and TROPOMI overpasses is about 4.2 min for the two cloud shadow band cases. For fast moving clouds this may give a shift in cloud and cloud shadow locations. For the two cloud band shadow cases discussed we investigated both ERA5 wind data and Spinning Enhanced Visible and InfraRed Imager (SEVIRI) RGB images. The SEVIRI images have a time resolution of 15 minutes and clearly show a southward movement of the cloud bands. The spatial resolution of SEVIRI together with possible cloud development make it challenging to precisely determine the speed of the cloud movement. We, however, estimate it to be on the order of 10-15 m/s in the southward direction perpendicular to the cloud shadow band. The ERA5 data have a large eastward component at the altitudes of the two cloud bands. For the 30 December 2019 case there is a much smaller southward component of about 10 m/s in agreement with the SEVIRI images. Surprisingly, for the 24 March 2019 case, the ERA5 data have a northward component of about 10 m/s, which is in disagreement with the SEVIRI observations. Trusting the SEVIRI images we find that the cloud mask and cloud shadow mask have shifted between 2.5 and 3.75 km perpendicular to the cloud shadow band between the TROPOMI and VIIRS overpasses. This is about the TROPOMI pixel size in this direction. For the 24 March 2019 case the cloud shadow band covers 1-2 TROPOMI pixels and it covers 2-4 TROPOMI pixels for the 30 December 2019 case. The cloud shadow band first viewed by VIIRS may thus be shifted southward when TROPOMI passes over. For the same geolocation, TROPOMI may thus view a smaller part of the cloud shadow band than VIIRS and hence be less affected by the cloud shadow. In Figs. 11 and 12 we average over the TROPOMI pixels identified to be affected by cloud shadow according to the VIIRS cloud shadow mask. Despite a possible reduction in the cloud shadow viewed by TROPOMI, a decrease is seen in the $NO_2$ TVCD for these pixels. We note that the cloud shift may in principle be corrected for using for example ERA5 data. However, as reported above, we find that SEVIRI and ERA5 data give different results with respect to cloud movement.

If it is assumed that the clouds are the main reason for the variations in the $NO_2$ TVCD over the cloud shadow bands, then these cases are examples of how cloud shadows give underestimates of $NO_2$ TVCD, in agreement with the theoretical idealized box cloud results presented by Emde et al. (2021) and Yu et al. (2021).

#### 4.2.2 Cloud effects for general cases

As shown in Fig. 9 for the synthetic data, the $NO_2$ bias may be on the order of tens of percent and are largest for large solar zenith angles. While the true $NO_2$ TVCD is known for the synthetic data, for the observational data the true $NO_2$ TVCD unaffected by clouds is not known. In order to have a reference or true $NO_2$ TVCD to compare with, we look at neighbour
pixels in a 3×3 pixel matrix where the pixel of interest is in the centre. The true $NO_2$ TVCD is then taken to be the average of cloudfree neighbours with $NO_2$ retrieval quality value > 0.95. Obviously this choice of true $NO_2$ TVCD has its problems including that neighbours may also be affected by clouds and that $NO_2$ TVCD may have large spatially gradients, see for example Fig. 2b which shows the $NO_2$ TVCD, and Fig. 2c which shows the percentage difference of the $NO_2$ TVCD to the area median $NO_2$ TVCD.

For the months of October 2018 and March 2019 when the solar zenith angle is between 50-60° for the study region (covering approximately Germany, the Netherlands and parts of other surrounding countries, see Introduction), we looked for $NO_2$ TVCD biases due to clouds. For October 2018 and March 2019 this amounted to a total of 1,023,081 TROPOMI pixels and 45,926,808 VIIRS pixels. To quantify possible cloud shadow effects and factors that impact it, TROPOMI pixels with $NO_2$ retrieval data quality value >0.95 and cloud shadows were selected for further analysis. A $NO_2$ retrieval with the data quality
value >0.95 was reported for 367,584 (36%) of the pixels. The VIIRS cloud mask identified 70.7% of the VIIRS pixels to be cloudy, indicating that clouds were the main reason for reducing the $NO_2$ retrieval quality for the majority of the TROPOMI pixels. Of the 367,584 pixels with high $NO_2$ retrieval data quality, a total of 129,180 (35%) were affected by cloud shadows according to the VIIRS cloud shadow product. Of the 45,926,808 VIIRS pixels 1,3438,968 (29.3%) were cloud free. Of these cloud free VIIRS pixels 17.8% contained cloud shadows. This number is lower than the number of TROPOMI pixels affected
by cloud shadows as is to be expected due to the higher spatial resolution of VIIRS. Note that this number pertains to months of the year where we expect cloud shadow effects to be large due to the large solar zenith angles. For months of the year with smaller solar zenith angles this number may well be smaller. The pixels affected by cloud shadows were further analysed to understand which parameters that have the largest impact on the $NO_2$ TVCD. In order to have a reference $NO_2$ TVCD to compare the centre pixel value with, only centre pixels which have one or more cloudfree neighbours with $NO_2$ retrieval
quality value > 0.95, were included. This restriction reduced the number of TROPOMI pixels to be analysed from 129,180 to 39,011. For these pixels the difference between the centre pixel $NO_2$ TVCD and the average of the $NO_2$ TVCD in the cloud free neighbours ($\Delta NO_2$) is shown as a function of cloud height maximum of neighbour pixels in Fig. 13. For this data subset clouds are mostly found at high (about 10,000 m) and low (about 1,900 m) altitudes, with relatively more clouds at the higher altitudes. This is qualitatively in agreement with Noel et al. (2018). They reported cloud fractions as estimated from a lidar
on board the International Space Station. For Europe (their Fig. 5b) they report a similar cloud vertical behaviour, albeit for a different time of year (July, June and August). The subset of TROPOMI pixels presented in Fig. 13 shows both high and low $NO_2$ TVCD biases with a median $NO_2$ TVCD bias of -1.4% and maximum of the Gaussian kernel density probability density function estimate maximum of -0.7%. Thus, for this subset of TROPOMI pixels no signifcant cloud shadow effect is visible in the $NO_2$ TVCD. As stated above, no true observational $NO_2$ TVCD is available as for the synthetic data. Furthermore, clouds

**Table 1.** Binning of parameters used to quantify the cloud shadow effect.

| Parameter | Bin borders |
|---|---|
| Maximum cloud height of neighbour pixels (NCH) | 0, 3000, 9000, 20000 |
| Maximum slant cloud optical thickness of neighbour pixels (SCOT) | 0, 3, 5, 7, 10, 15, 300 |

and thus cloud shadows may have moved between VIIRS and TROPOMO overpasses. This may be reasons for the lack of a clear cloud shadow effect in this data subset.

This subset of data were further binned according to the maximum cloud height of neighbour pixels and the maximum slant cloud optical thickness (SCOT) of neighbour pixels as given in Table 1. The maximum slant cloud optical thickness is the neighbour pixel maximum cloud optical thickness divided by the cosine of the solar zenith angle for that pixel. Furthermore, only TROPOMI pixels where the maximum slant cloud optical thickness and maximum cloud top height are in the same neighbour pixel were included. This reduced the data set to 18,029 pixels.

The $NO_2$ TVCD bias density as a function of the cloud shadow fraction for the SCOT bins and maximum cloud heights is shown in Fig. 14. As the cloud height increases (from bottom row to top row in Fig. 14) the cloud shadow fraction increases because generally the cloud shadow within a pixel geometrically increases with cloud height when the cloud is larger than the pixel size. For the low clouds the median $NO_2$ TVCD bias varies between -1.2 to 0.2% for SCOT<15. For SCOT>15 the median $NO_2$ TVCD bias is -2.9%. For the medium height clouds the median $NO_2$ TVCD bias for SCOT>15 increase in magnitude to -5.8%. For smaller SCOT the median $NO_2$ TVCD bias is negative and varies between -0.8 and -3.4%. For the high clouds the median $NO_2$ TVCD bias is between -0.3 and 1.6% for SCOT<7. For larger SCOT the median $NO_2$ TVCD bias is -0.9, -8.7 and -15.1% (7<SCOT<10, 10<SCOT<15, 15<SCOT<300).

For all cloud heights the median $NO_2$ TVCD bias is negligible for SCOT<7. For low and medium clouds there is a negative bias for the largest SCOT. For the high clouds the bias is pronounced for SCOT>10. Thus, the median $NO_2$ TVCD bias increases with cloud height and with slant cloud optical thickness. It is noted that for individual pixels the bias may be larger and both positive and negative. We use a neighbour pixel as the true $NO_2$-value. This assumes that only the cloud shadow effect is the reason for the $NO_2$ TVCD bias. In reality there are horizontal gradients in the $NO_2$ TVCD for numerous other causes as well, including local emissions and wind transport differences. Also, as is discussed above for the cloud shadow band cases, the difference in overpass time between VIIRS and TROPOMO, may give differences in the clouds and cloud shadows viewed by the two instruments.

Finally it is noted that for the 129,180 pixels which were affected by cloud shadows, the number of pixels with cloud free neighbours where $|\Delta NO_2|>20\%$ is 45.7%. Under the assumption that this number is applicable to all TROPOMI pixels affected by cloud shadows, about 16% of TROPOMI pixels for which $NO_2$ TVCD retrievals with a high quality value were made, may be impacted by cloud effects larger than 20% for solar zenith angles between 50-60°. Of these about half of the $NO_2$ TVCD are overestimated and half underestimated. The underestimate is clearly linked to cloud shadow effects. The

overestimate may be due to in-scattering or horizontal gradients in $NO_2$ concentrations and thus wrong cloud shadow free $NO_2$ TVCD true value. These two processes may, however, not be distinguished in the observed data set.

## 5 Conclusions

In this study we have investigated the impact of 3D clouds on $NO_2$ TVCD retrievals from UV-VIS sounders. Both synthetic and observational data have been used to identify and quantify possible biases in $NO_2$ TVCD retrievals. The synthetic data were based on high-resolution LES results which were input to the MYSTIC 3D radiative transfer model. The simulated visible spectra for low-earth orbiting and geostationary geometries were analysed with standard $NO_2$ retrieval methods including operational cloud corrections. Profiles of $NO_2$ for polluted conditions, with increased $NO_2$ in the lower atmosphere below cloud tops, were considered as cloud shadow effects are not important for background $NO_2$ conditions where the amount of $NO_2$ below the cloud top is relatively small compared to the total column. For the observational data the $NO_2$ products from TROPOMI were used while VIIRS provided high spatial resolution cloud data. Both single cases and overall statistics were calculated. The main findings are:

- The following metrics were identified as being the most important in identifying 3D cloud impacts on $NO_2$ retrievals: cloud shadow fraction, cloud top height, cloud optical thickness, solar zenith and viewing angles.

- Analysis of the synthetic data show that for low-earth and geostationary orbit geometries, 89 and 93%, respectively, of the retrieved $NO_2$ TVCD are within 10% of the actual column for small solar zenith angles. For large solar zenith angles the numbers decrease to 53 and 61%.

- The synthetic data shows that in general the $NO_2$ TVCD bias is slightly larger for low-earth orbiting geometries than for geostationary geometries. This is due to differences in viewing geometry where, for the mid-latitude targets studied here, the sun-target-detector geometry overall sees fewer cloud shadows for geostationary geometry.

- For a solar zenith angle less than about $40°$ the synthetic data show that the $NO_2$ TVCD bias is below 10%. For larger solar zenith angles both synthetic and observational data show $NO_2$ TVCD bias on the order of tens of %.

- For clearly identified cloud shadow bands in the observational data, the $NO_2$ TVCD appears low-biased when the cloud shadow fraction $> 0.0$ compared to when the cloud shadow fraction is zero. If it is assumed that the clouds are the main reason for the variations in the $NO_2$ TVCD over the cloud shadow band, i.e. the $NO_2$ field is assumed to be horizontally homogeneous, then these cloud shadow band cases are examples of how cloud shadows give underestimates of $NO_2$ TVCD, in agreement with the theoretical findings.

The above conclusions suggest that further work is required on 3D cloud radiative impacts. For this future work the following topics may be considered:

- As shown in this study, 3D radiative simulation of high-resolution satellite instrument spectra with realistic 3D cloud input is fully achievable. However, to cover all possible atmospheric situations on Earth, more high-spatial resolution

LES results may be needed. Furthermore, 3D RT modelling is demanding on computer resources and requires careful interpretation and analysis of the output. It is expected that such modelling efforts may prove useful not only for trace gas retrieval algorithm studies, but also for cloud detection and cloud microphysics retrievals and more.

– For future missions, including the Meteosat Third Generation (MTG) Flexible Combined Imager (FCI) and the Meteorological Operational Satellite Second Generation METimage (formerly known as VII-Visible and Infrared Imager), a cloud shadow product is needed for assessing and mitigating 3D cloud impacts. Experience with the VIIRS cloud shadow product (large changes between versions[1]) suggests that independent verification with ground measurements may be of use. Such validation is non-trivial and possibly require new experimental approaches for measurements of both cloud shape and trace gas spatial variation at sub-pixel resolution.

*Code availability.* The libRadtran software used for the radiative transfer simulations is available from www.libradtran.org. The QDOAS software for DOAS retrieval of trace gases is available from uv-vis.aeronomie.be/software/QDOAS.

*Data availability.* VIIRS data were accessed through the NOAA Comprehensive Large Array-Data Stewardship System (CLASS, https://www.class.noaa.gov). TROPOMI data were downloadded from https://s5phub.copernicus.eu/

**Appendix A: Absorbing aerosol index**

Recently Kooreman et al. (2020) identified and explained AAI large scale effects such as the cloud bow, sunglint and viewing zenith angle dependence. In addition they reported small-scale negative AAI values in partly cloudy areas and attributed this to 3D cloud structures casting shadow, but left the in-depth analysis for a future study,

As cloud shadow impact both AAI and $NO_2$ TVCD retrievals it is of interest to investigate possible relationships between the AAI and the cloud shadow fraction and the $NO_2$ TVCD. For the partly cloudy scene in Fig. 10, the TROPOMI AAI from 380 and 340 nm is shown in Fig. A1a. Overall the AAI is negative indicating the absence of absorbing aerosol, but the presence of scattering particles. The behaviour of clouds on AAI is complex. For effective cloud fraction between 30-50% (5-30%) for thick (thin) clouds Penning de Vries et al. (2009b) reported negative AAI while for large cloud fractions high, thick clouds may cause positive AAI. The increase in AAI from scattered clouds to complete cloud cover may be seen when comparing Fig. 10d and Fig. A1a. For pixels with cloud shadows, the AAI does not vary with the cloud shadow fraction as shown in Fig. A1b. It is noted that the $NO_2$ TVCD varies considerably for these pixels, Fig. A1c. Thus, there appears to be no obvious dependence between the AAI and the $NO_2$ TVCD.

---

[1]The VIIRS L2 product changed version from v1r1 to v1r2 between 13 and 14 Aug 2018, see https://www.star.nesdis.noaa.gov/jpss/documents/AMM/N20/Cloud_CBH_Provisional.pdf. Large changes in the cloud shadow product was seen between versions with v1r1 given unrealistic large number of pixels with cloud shadow. Realistic numbers were found with v1r2.

*Author contributions.* AK: Conceptualization, Methodology, Formal analysis, Writing - Original Draft. CE: Conceptualization, Methodology, Formal analysis (3D radiative transfer simulations). HY: Conceptualization, Methodology, Formal analysis ($NO_2$ retrievals) MvR, KS, BV and BM: Conceptualization, Methodology. All: Writing - Review & Editing

*Competing interests.* The authors declare that no competing interests are present.

5   *Acknowledgements.* This work was funded by ESA (3DCATS project 4000124890/18/NL/FF/gp). We thank Nikolaos Evangeliou for helping with ERA5 data. We would also like to thank the two anonymous referees for their comments.

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

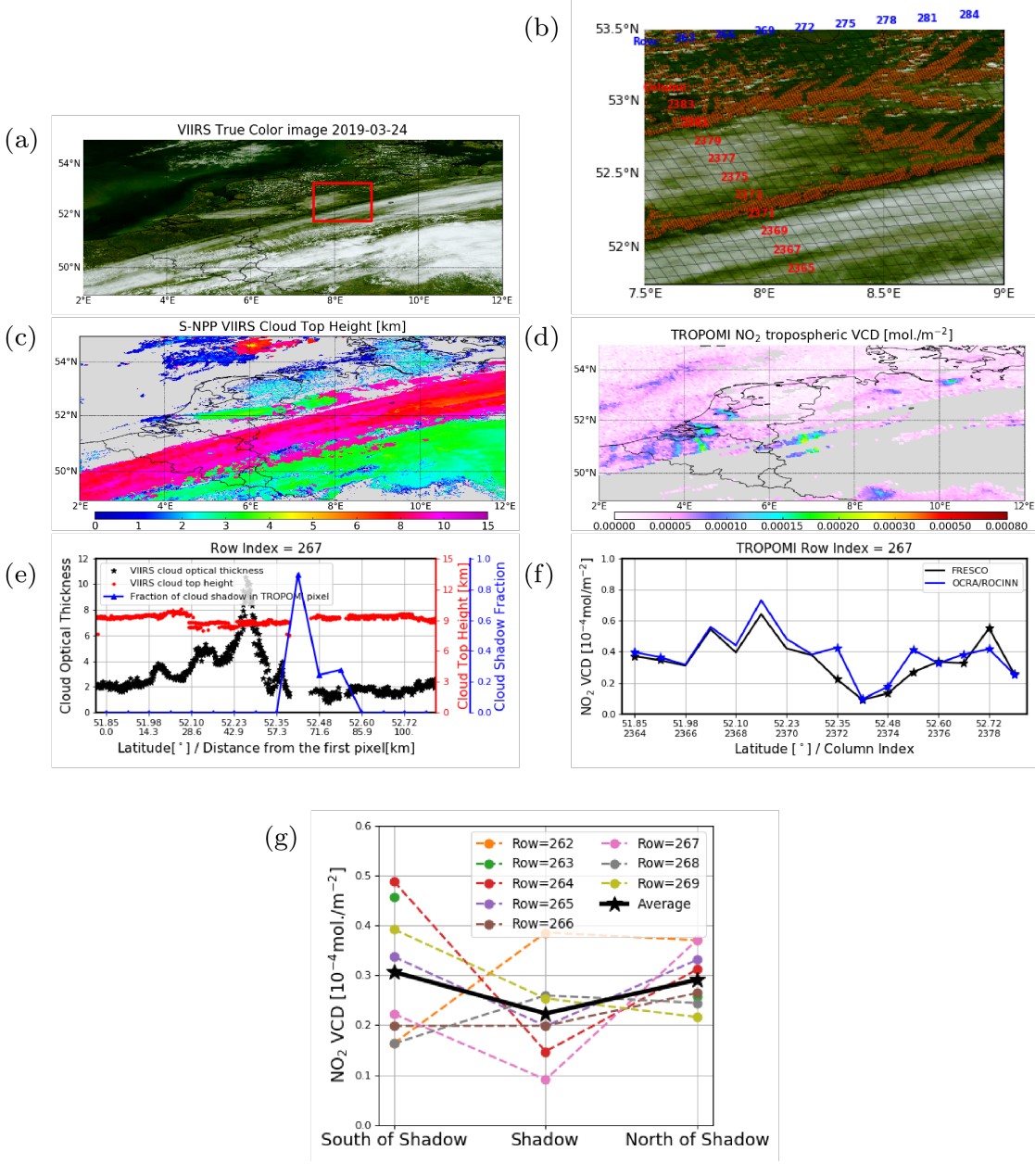

**Figure 11.** Another view of the cloud shadow band example in Fig. 10. (a) VIIRS RGB image; (b) zoom-in of cloud shadow band with TROPOMI footprint, red marks indicate pixels with cloud shadow; (c) the VIIRS cloud top height; (d) the TROPOMI $NO_2$ TVCD; (e) the VIIRS cloud optical thickness, cloud top height and cloud shadow fraction for TROPOMI row 267 (the row localisation is shown in panel b); (f) the $NO_2$ TVCD using FRESCO and OCRA/ROCINN cloud correction algorithms, as a function of latitude. The star marks represent pixels with $CF_w < 50\%$; (g) averaged $NO_2$ TVCD for pixels south of the shadow, in the shadow and north of the shadow for rows 262-269 and the average of these. See text for further details.

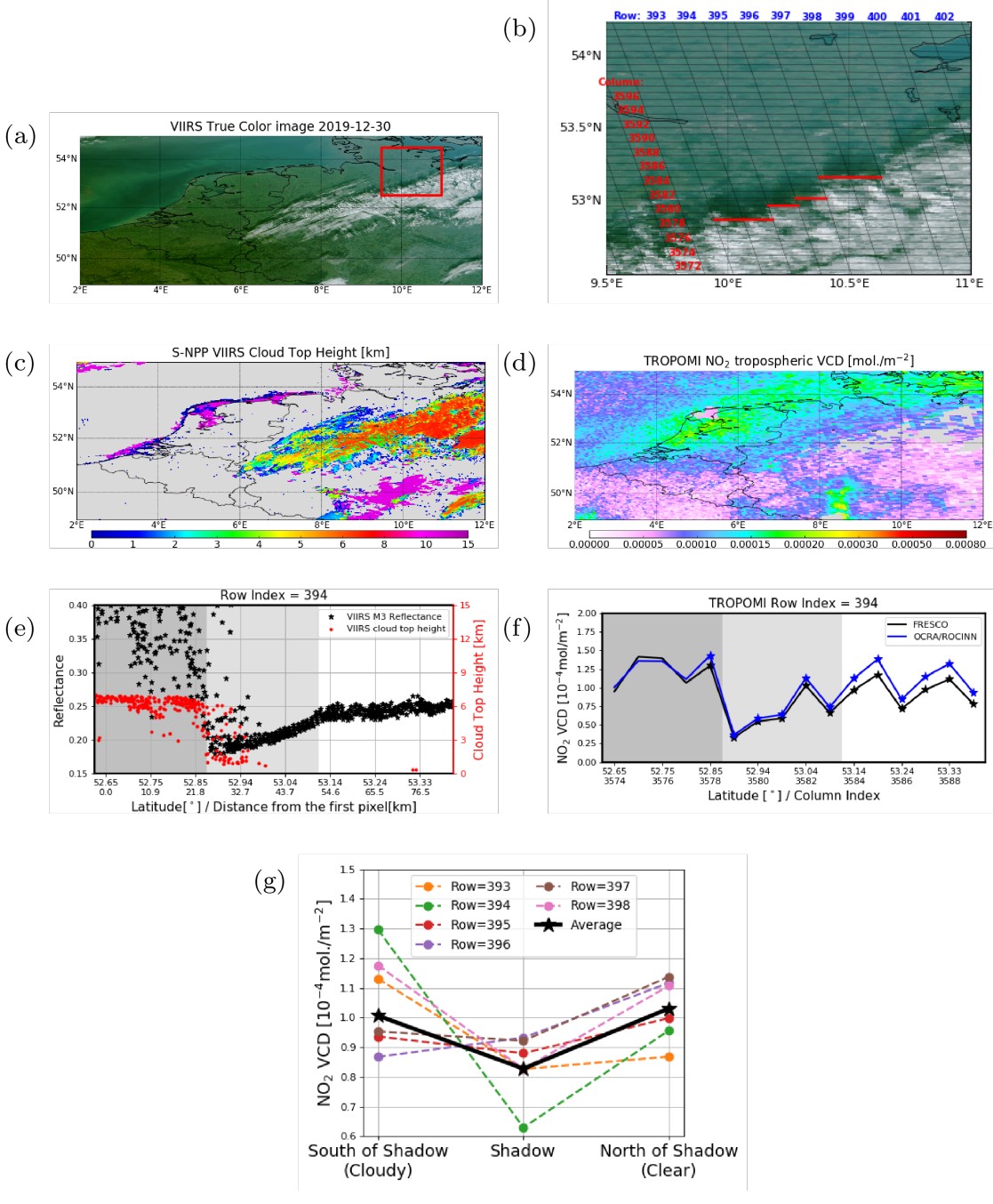

**Figure 12.** Similar to Fig. 11, but data for 30 December 2019. Furthermore, in (b) the red lines indicate the cloud edge in along-track direction for rows 393 to 398; in (e) no cloud shadow nor cloud optical thickness information is included, instead the VIIRS M3 reflectance is plotted; in (e)-(f) cloud, shadow and clear regions are identified as dark gray/light gray/white regions.

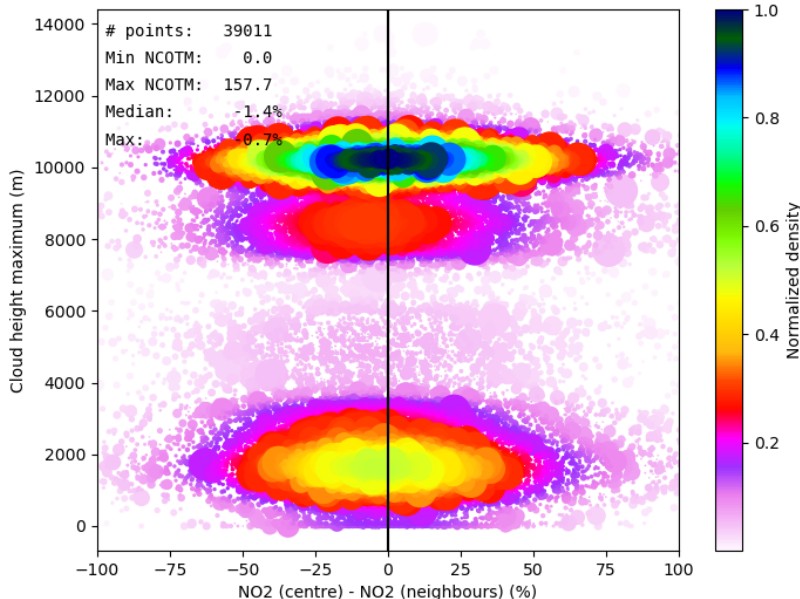

**Figure 13.** The difference between the centre pixel NO$_2$ TVCD and the average of the NO$_2$ TVCD in the cloudfree neighbours as a function of cloud height maximum of neighbour pixels. The median of the difference is given in the legend together with the maximum of a Gaussian kernel-density probability density function estimate of the data points. The size of the data points illustrates the maximum cloud optical thickness in neighbour pixels (NCOTM). It varies between the minimum and maximum NCOTM values given in the legend.

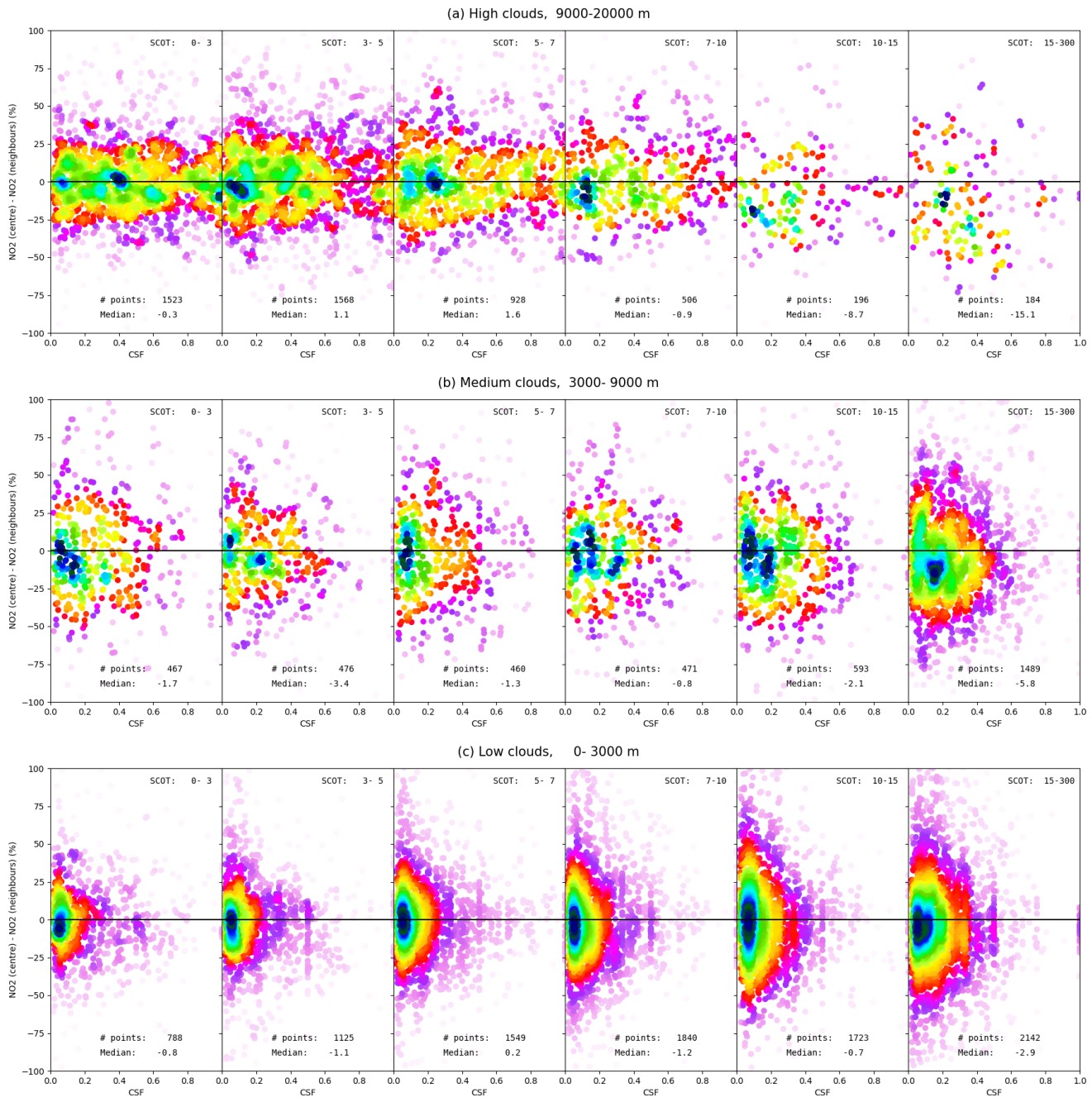

**Figure 14.** The NO$_2$-bias density as a function of the cloud shadow fraction. The data are binned into slant cloud optical thickness bins for high (a), medium (b) and low (c) clouds. See text for further details.

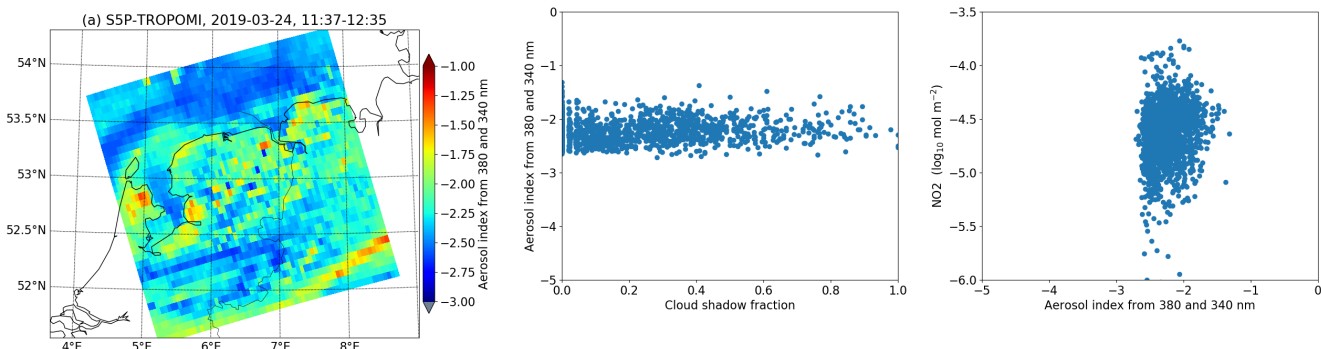

**Figure A1.** (a) The TROPOMI aerosol index. (b) The TROPOMI aerosol index versus the cloud shadow fraction. (c) The TROPOMI aerosol index versus the NO$_2$ TVCD. For (b) and (c) only data points where the NO$_2$ TVCD data quality flag $> 0.95$ and the AAI data quality flag $> 0.5$, are included. All data from 24 March 2019.