# Peer review of "Impact of 3D Cloud Structures on the Atmospheric Trace Gas Products from UV-VIS Sounders - Part III: bias estimate using synthetic and observational data"

_Atmospheric Measurement Techniques, 2021_

## Referee Comment (RC1)

Review of Impact of 3D Cloud Structures on the Atmospheric Trace Gas Products from UV-VIS Sounders - Part III: bias estimate using synthetic and observational data by Arve Kylling et al

This paper is one of a set of three papers that discusses a) a publicly available synthetic dataset of 3D radiances, b) the sensitivity of vertical column density NO2 retrieval errors near box-clouds and observations, and c) 3D cloud biases and metrics. The reviewed paper is part b) of the full set of papers.

The paper is well written, concise, clear, and informative. The paper is well organized, and the figures are thoughtfully chosen and carefully presented. The text lines on page 8 (lines 13-16), page 13 (lines 10-13), and the Conclusions (pages 21-24) are especially well written. Since there are relatively few papers which discuss 3D radiative transfer and its impact on satellite retrievals, the paper is fairly unique.

The paper should be published after minor revision.

General comments

On page 9, lines 15-19. How far from clouds must one go to have the AMF bias to be less than 20%? This would be a useful "rule of thumb" approximate value for the user community to learn and remember. The authors have the opportunity to educate the general research community in regard to the general quantitative importance of 3D radiative transfer effects, and its impact on NO2 retrievals, and I encourage the authors to do so in this paper.

On page 23, line 2, the authors state that "cloud shadow effects are not important for background NO2 conditions.". Please clarify why this is the case.

Specific comments

Page 2, lines 31-32. Rephrase to "The retrieved NO2 using standard 1D algorithms was compared to the input to the 3D radiative transfer simulations and possible 3D radiative effects were identified and quantified."

Page3, line 24. Rephrase to "Note that each simulated sensor pixel includes 36 cloud pixels, hence the simulations include"

Page 4, line 11. Rephrase to "Combining the sun-sensor geometries .."

Page 9, line 1. Rephrase to "The bias decreases to 0% when the CFw is between 1-3%."

Page 17, line 2. Rephrase to "satisfied this criteria and therefore no data is shown"

Page 21, lines 5-6. Rephrase to "the cloud shadow fraction increases because generally the cloud shadow within a pixel geometrically increases with cloud height".

Criteria

1. Does the paper address relevant scientific questions within the scope of AMT? Yes

2. Does the paper present novel concepts, ideas, tools, or data? There are relatively few papers than discuss 3D radiative transfer, so the paper is fairly unique.

3. Are substantial conclusions reached? Yes

4. Are the scientific methods and assumptions valid and clearly outlined? Yes

5. Are the results sufficient to support the interpretations and conclusions? Yes

6. Is the description of experiments and calculations sufficiently complete and precise to allow their reproduction by fellow scientists (traceability of results)? Yes

7. Do the authors give proper credit to related work and clearly indicate their own new/original contribution? Yes

8. Does the title clearly reflect the contents of the paper? Yes

9. Does the abstract provide a concise and complete summary? Yes

10. Is the overall presentation well structured and clear? Yes

11. Is the language fluent and precise? Yes

12. Are mathematical formulae, symbols, abbreviations, and units correctly defined and used? Yes

13. Should any parts of the paper (text, formulae, figures, tables) be clarified, reduced, combined, or eliminated? The General Comments section above points out a few places (on pages 9 and 23) where clarifications are suggested.

14. Are the number and quality of references appropriate? Yes

15. Is the amount and quality of supplementary material appropriate? Yes

---

## Referee Comment (RC2)

Review of Kylling et al. 2021: Impact of 3D Cloud Structures on the Atmospheric Trace Gas Products from UV-VIS Sounders -- Part III: bias estimate using synthetic and observational data

General comments:

The authors did a comprehensive analysis of 3D cloud effects on synthetic and observational data of NO2 concentrations. A lot of simulations were performed regarding the NO2 and AMF bias due to 3D cloud effects, and the biases as functions of physical parameters such as illumination geometry, viewing geometry and surface albedo were explored.

The presentation of the results needs an improvement. Most figures are difficult to read: they contain too much information (Fig. 9 and 10) and/or are too small (e.g., Fig. 12) and/or a different type of graph could be more suitable (why are the dots in Fig. 5, 9 and 10 connected by lines?) and/or could better be skipped (Fig. 9.b and 10.b) (see detailed comments). Please avoid legends overlapping the graphs (e.g., in Fig 9 and 10). The description of some figures is sometimes insufficient (Fig 2. for example, does not have any explanation in the text), or is unclear (e.g., what is the added value of the skewness etc. in Figs 9.b and 10.b to your conclusions)?

The VIIRS cloud mask and cloud shadow mask are used, but are not reliable for cloud and cloud shadow identification in the analysis of cloud shadow signatures in TROPOMI data. This is because cloud shadows are small-scale features (1 or a few TROPOMI pixels). The overpass measurement time difference between TROPOMI and VIIRS, which is a couple of minutes, is enough to move clouds at least 1 TROPOMI pixel (see ESA-ATMOS symposium oral presentation by Trees et al., 2021 https://atmos2021.esa.int/iframe-agenda/files/Contribution\_171\_final\_extabs.pdf). This is particularly true for clouds that produce cloud shadows visible from space, because those shadow-producing clouds must be located at high altitudes where the wind speeds are relatively high. Additionally, near the cloud edges cloud evolution (i.e., cloud shape change) occurs. Cloud shadows should therefore be identified using measurements taken at the TROPOMI measurement time.

In Section 4.2.1, zoomed in areas are considered (a few pixel rows) of only two cases, while the spatial natural variability of the NO2 VCD is actually very high (see e.g. your Figs. 2.c, 12.d and 13.d). In order to make sure that the observations are caused by 3D cloud effects instead of the natural variability, statistics of more observations are needed. In Section 4.2.2 shadowed pixels are compared with shadow-free neighboring pixels, but the cloud movement and evolution (see former paragraph) could consistently result in the situation where the actual shadows are located inside the neighboring pixels, while the identified shadow pixels are in fact shadow-free. Consistently confusing the shadowed pixels with non-shadowed pixels may result in false conclusions about the observed shadow induced NO2 signatures.

The results shown in Section 4.2 are highly scattered (indeed no clear relation between TROPOMI NO2 and VIIRS shadow fraction can be derived from Fig. 11.g, Fig. 14 and Fig 15). However, in the conclusions and abstract, it is written that NO2 appears low-biased in observations. Considering the scattered results, together with the questionable approach that was followed (see former paragraphs), no reliable conclusions can be drawn from this analysis about the observed NO2 bias dependence on shadows. Therefore, I suggest removing Section 4.2 and Appendix A from this paper and limit the analysis to only the modelling part (Section 4.1).

Detailed comments:

- Page 1, line 10-11: limit your conclusion to only synthetic data.
- Page 1, line 11-14: the low NO2 bias in the observations is not significant (see later comments below).
- Page 4, Fig 1.b and Fig 1.c: Please avoid using a white color for the cloud pixels, since white is also one of the colors in the color bar.
- Page 4, Fig 1.c: The cloud shadow index is 1 below the clouds, but this would be invisible from space. I suggest removing the cloud pixels in this plot (such as in Fig. 1.b), such that the cloud shadows become visible.
- Page 5, Fig 2.a: It is difficult to see the colors of the RGB. Can you try to increase the brightness and/or enhance the colors?
- Page 5: Fig 2: Please increase the size of the images. Discuss every subfigure when you introduce Fig. 2 in the text or remove the subfigure.
- Page 6, line 13: Why do you mention FRESCO here? Are you using FRESCO for your analysis? This is not yet clear for the reader at this point.
- Page 6, line 28: You do not consider ocean cases. On Page 3, line 18, you mention that your focus is on Europe and different cloud types, and that therefore the results are expected to be general and applicable elsewhere. Would your results also be representative over ocean? And over desert, or over snow/ice?
- Page 7, line 1-3: 'they compare the spectral procedure', 'and found the latter to be far superior'. Could you rephrase this? What is 'the spectral procedure'? What do you mean by 'far superior'?
- Page 7: what are the accuracies of the VIIRS cloud mask and VIIRS shadow mask (regardless of the mapping to the TROPOMI grid)?
- Page 7, line 16: From this explanation, it seems that the Cloud Shadow Index (CSI) also indicates fully cloudy pixels (as also shown in Fig. 1.c). Why is this the case? Wouldn't excluding cloudy pixels be better for the analysis of shadows which uses the CSI, such as Fig. 3? Or did you indeed apply a cloud filter? Please explain in this subsection, this is not clear for the reader at this point.
- Page 7, line 29-30: Why are the "" used here? Is this a citation? This is not a proper explanation of the AAI. Please rephrase and add a reference to de Graaf et al. (2005): https://agupubs.onlinelibrary.wiley.com/doi/full/10.1029/2004JD005178. For example: 'The AAI is a measure of the UV color of a cloud-, aerosol- and shadow-free 1-D atmosphere-surface model with respect to the measured UV color (de Graaf et al., 2005). When absorbing aerosols are present, the AAI tends to be positive, while the AAI is approximately zero or negative in the presence of clouds (see e.g. Kooreman et al., 2020; Penning de Vries et al., 2009).'
- Page 8, Fig 3: 'Please remove cloud pixels from your analysis, and if you did (already), please mention this in the caption of the figure.'
- Page 8, line 14-16: "The cloud shadow impact ... respectively." Please rephrase, this sentence is difficult to read. Are the percentages you mention here average values for a CSI of 1?

- Page 8, line 16: "As the solar zenith angle increases a linear relationship appears ...". Is there really a linear relationship? The data looks scattered. Can you please quantify the linear relationship with the corresponding uncertainty?
- Page 8, line 20: "For both geometries the NO2 AMF is high ... is between 1-3%." Why? Can you physically explain these numbers here already? If not, please refer to this location in the paper explicitly when you can. For example, on page 12 line 11, you investigated the cause, and you can refer back: 'This explains the high AMF biased that we observed in Fig. 4a for CFw
- Page 14, line 1: '... also occur.' Can you refer to the figure(s) where this was shown?
- Page 14, line 1-2: '..., such as cloud top altitude and cloud optical thickness, are also of importance.' How did you come to this conclusion? Can you show this or refer to the figure where this has been shown?
- Page 14, line 5: How precisely is the cloud enhancement effect visible in Figs. 9 and 10? Please explain.
- Page 14, line 7: 'theta = 20 30 degrees'. How can the viewing zenith angle be observed in Fig. 9a?
- Page 14, line 12 to Page 15, line 4: What is the message of this paragraph? Do you mean that 3D cloud and cloud shadow effects are smaller than the NO2 retrieval uncertainty?
- Page 15, line 7-12: Should these sentences be part of Section 4.2.2 instead?
- Page 15, line 20-22: please rephrase: : "For a cloud shadow fraction ... standard deviation". Please add: "The scatter in Fig. 11.g is too large to draw conclusions about the dependence of NO2 on shadow fraction."
- Page 15, line 24-25: "Thus indicating that ... cloud cases." Please rephrase this sentence: what is the subject of this sentence?
- Page 16: can you please make the figures bigger?. Also, in Fig. 11.g, the lime green squares are not visible.
- Page 16, Fig. 11: can you explain the oscillatory pattern in the geometric cloud fraction (Fig. 11.d) and the cloud shadow fraction (Fig. 11.f) in the shadow band?
- Page 16, fig 11.g: how do you precisely define "CSF pixels"? Doesn't each pixel has a certain CSF? Please clarify this in the caption.
- Page 16, Fig 11.g: the variability is much larger than the differences between NO2 (all pixels) and NO2 (CSF pixels). No significant relation between NO2 and CSF can be identified with this figure. Please clarify this explicitly in the text.
- Page 17, line 4: "For the cloud shadow band the NO2 TVCD is on average reduced by 17%". Please add a sentence here explaining that only a few pixel rows are analyzed, while the NO2 natural spatial variation is actually very large (Fig. 12.d).
- Page 17, line 22-23: "All cases show that the NO2 TVCD in the cloud shadow is lower by 8-46% (average of 25%) compared with the NO2 TVCD around the shadow." What about pixel row 396? Pixel row 396 seems to have a higher NO2 TVCD in the shadow than south of shadow.

- Page 17, line 26-28: "If it is assumed that the clouds are the main reason for the variations in the NO2 TVCD over the cloud shadow bands, then these cases are examples of how cloud shadows give underestimates of NO2 TVCD, in agreement with the theoretical idealized box cloud results presented by Emde et al. (2021) and Yu et al. (2021)." I don't think you can conclude this, given the high spatial NO2 variability (Figs. 2.c, 12.d, 13.d), the limited number of cases and pixel rows that were analyzed, the high scatter of the NO2 bias as functions of shadow fraction (Fig. 11.g), and the questionable approach to mask clouds and shadows using VIIRS masks on the TROPOMI grid (due to the cloud movement and cloud evolution during the TROPOMI-VIIRS overpass time difference, the undiscussed VIIRS mask accuracy, and the oscillatory features in the geometric cloud fraction and shadow fraction in the shadow band (Fig.s 11d and 11f)).
- Page 17, line 34: TROPOMI processes 25 million pixels per day. Why do you use for October 2018 and March 2019 only 1023081 pixels? What is the study region precisely?
- Page 18, line 1: 35% of what precisely? 35% is a large percentage for cloud shadows, even in months where you expect cloud shadows. Can you please verify this number? How does this number relate to the overall cloud fraction of the data set?
- Page 18 and 19, Fig 12 and 13: Fig 12 and 13 are hard to read. Please make the figures bigger. Figure 12.b and 13.b are problematic: can you ensure that the cloud movement and evolution during the TROPOMI-VIIRS overpass time difference did not consistently affect the shadow identification?
- Page 18 and page 19, Fig.12.g and 13.g: only a couple of pixel rows are analyzed, and even within this small sample, the low NO2 bias is not consistent. For example, in Fig. 12.g, the NO2 TVCD is higher in the shadow than outside the shadow for rows 262 and 265.
- Page 20, line 10: "no true NO2 TVCD is available as for the synthetic data" -> do you mean "observational data"?
- Section 4.2.2 general comment: The results in Figs. 14 and 15 are highly scattered, and no clear negative NO2 bias from cloud shadows can be determined. This should be clear in the text, conclusions and abstract.
- Section 4.2.2 general comment: neighbor pixels in a 3x3 pixel matrix where used, and the true NO2 TVCD is then taken to be the average of the cloud-free neighbors. Cloud movement and evolution during the measurements time difference of TROPMI and VIIRS could consistently result in the situation where the actual shadows are located inside the neighboring pixels, while the identified shadow pixels are in fact shadow-free.
- Page 23, line 15-19: "For clearly identified cloud shadow bands ... with the theoretical findings." Why can you assume that the clouds are the main reason for the spatial NO2 variations / assume that the NO2 background is horizontally homogeneous?
- Page 23, line 20-21: "For a solar zenith ... to be impacted by cloud effects larger than 20%". Where did you show this? Also, please mention that the data is very scattered and comment on the uncertainty of your conclusions.

• Page 24, line 1: You mention that there are "large changes between versions" of the VIIRS cloud shadow product. Could you elaborate on that? What is the accuracy of the VIIRS cloud shadow product itself (regardless of the mapping onto the TROPOMI grid)?

**Appendix A**

- Page 24, line 12: "As cloud shadow impact NO2 TVCD retrievals, ..." Do you mean instead: "As cloud shadow impact both AAI and NO2 retrievals, ..."?
- Page 24, line 15: "Indeed, over land the AAI is more negative over cloudy pixels, compare Fig. 11d and Fig. A1a". -> This seems not really to be the case when looking at Fig. 11d and Fig. A1a: the large cloud deck between 52 deg N and 52.5 deg N does not give more negative AAI. Clouds do not always decrease the AAI, they usually just don't *increase* the AAI (see e.g. Penning de Vries et al., 2009).
- Page 25, line 2: "..., while the NO2 TVCD shows some dependency on cloud shadow fraction, Fig. 11g." -> Please remove this part, the dependency on cloud shadow fraction from Fig. 11g is insignificant given the high variability.

---

## Author Comment (AC1)

**Response to interactive comments from Referee #1**

Below the comments from Referee #1 are given in italic font. Our responses to the comments are shown in roman font.

**General comments**

- *On page 9, lines 15-19. How far from clouds must one go to have the AMF bias to be less than 20%? This would be a useful rule of thumb approximate value for the user community to learn and remember. The authors have the opportunity to educate the general research community in regard to the general quantitative importance of 3D radiative transfer effects, and its impact on NO2 retrievals, and I encourage the authors to do so in this paper.*

  This is an interesting question that unfortunately does not have a simple answer, but depends on solar zenith angle, cloud top height, pixel size and more. To address the comment we have added the following:

  > For a solar zenith angle around 40° most pixels will have an AMF bias below 20% if the distance to the cloud edge is more than about 10 km. For a solar zenith angle of 60° this distance increase to about 20 km. The distance depends on a number of factors such as cloud top height, cloud optical depth and surface albedo. This is further discussed and quantified for box clouds in the accompanying paper by Yu et al. (2021).

- *On page 23, line 2, the authors state that cloud shadow effects are not important for background NO2 conditions.. Please clarify why this is the case.*

  We have clarified this by changing the sentence to:

  > Profiles of $NO_2$ for polluted conditions, with increased $NO_2$ in the lower atmosphere below cloud tops, were considered as cloud shadow effects are not important for background $NO_2$ conditions where the amount of $NO_2$ below the cloud top is relatively small compared to the total column.

**Specific comments**

- *Page 2, lines 31-32. Rephrase to The retrieved NO2 using standard 1D algorithms was compared to the input to the 3D radiative transfer simulations and possible 3D radiative effects were identified and quantified.*

  The sentence has been changed as suggested.

- *Page3, line 24. Rephrase to Note that each simulated sensor pixel includes 36 cloud pixels, hence the simulations include*

  The sentence has been changed as suggested.

- *Page 4, line 11. Rephrase to Combining the sun-sensor geometries ..*

  Correction made as suggested.

- *Page 9, line 1. Rephrase to The bias decreases to 0% when the CFw is between 1-3%.*

  Correction made as suggested.

- *Page 17, line 2. Rephrase to satisfied this criteria and therefore no data is shown*

  Correction made as suggested.

- *Page 21, lines 5-6. Rephrase to the cloud shadow fraction increases because generally the cloud shadow within a pixel geometrically increases with cloud height.*

  Corrections made as suggested.

**Bibliography**

Yu, H., Emde, C., Kylling, A., Veihelmann, B., Mayer, B., Stebel, K., and van Roozendael, M.: Impact of 3D Cloud Structures on the Atmospheric Trace Gas Products from UV-VIS Sounders - Part II: impact on NO 2 retrieval and mitigation strategies, Atmospheric Measurement Techniques, submitted, 2021.

---

## Author Comment (AC2)

**Response to interactive comments from Referee #2**

Below the comments from Referee #2 are given in italic font. Our responses to the comments are shown in roman font.

**General comments**

- *The presentation of the results needs an improvement. Most figures are difficult to read: they contain too much information (Fig. 9 and 10) and/or are too small (e.g., Fig. 12) and/or a different type of graph could be more suitable (why are the dots in Fig. 5, 9 and 10 connected by lines?) and/or could better be skipped (Fig. 9.b and 10.b) (see detailed comments). Please avoid legends overlapping the graphs (e.g., in Fig 9 and 10). The description of some figures is sometimes insufficient (Fig 2. for example, does not have any explanation in the text), or is unclear (e.g., what is the added value of the skewness etc. in Figs 9.b and 10.b to your conclusions)?*

  An explanation of Fig. 2 have been added to the text. The other points have been answered in specific comments 23 and 26 below.

- *The VIIRS cloud mask and cloud shadow mask are used, but are not reliable for cloud and cloud shadow identification in the analysis of cloud shadow signatures in TROPOMI data. This is because cloud shadows are small-scale features (1 or a few TROPOMI pixels). The overpass measurement time difference between TROPOMI and VIIRS, which is a couple of minutes, is enough to move clouds at least 1 TROPOMI pixel (see ESA-ATMOS symposium oral presentation by Trees et al., 2021 $https://atmos2021.esa.int/iframe-agenda/files/Contribution\_171\_final\_extabs.pdf$). This is particularly true for clouds that produce cloud shadows visible from space, because those shadow-producing clouds must be located at high altitudes where the wind speeds are relatively high. Additionally, near the cloud edges cloud evolution (i.e., cloud shape change) occurs. Cloud shadows should therefore be identified using measurements taken at the TROPOMI measurement time.*

  The point raised have been answered in specific comment 47 below.

- *In Section 4.2.1, zoomed in areas are considered (a few pixel rows) of only two cases, while the spatial natural variability of the NO2 VCD is actually very high (see e.g. your Figs. 2.c, 12.d and 13.d). In order to make sure that the observations are caused by 3D cloud effects instead of the natural variability, statistics of more observations are needed. In Section 4.2.2 shadowed pixels are compared with shadow-free neighboring pixels, but the cloud movement and evolution (see former paragraph) could consistently result in the situation where the actual shadows are located inside the neighboring pixels, while the identified shadow pixels are in fact shadow-free. Consistently confusing the shadowed pixels with non-shadowed pixels may result in false conclusions about the observed shadow induced NO2 signatures.*

  The points raised have been answered in specific comment 47 below.

- *The results shown in Section 4.2 are highly scattered (indeed no clear relation between TROPOMI NO2 and VIIRS shadow fraction can be derived from Fig. 11.g, Fig. 14 and Fig 15). However, in the conclusions and abstract, it is written that NO2 appears low-biased in observations. Considering the scattered results, together with the questionable approach that was followed (see former paragraphs), no reliable conclusions can be drawn from this analysis about the observed NO2 bias dependence on shadows. Therefore, I suggest removing Section 4.2 and Appendix A from this paper and limit the analysis to only the modelling part (Section 4.1).*

We disagree with the reviewer that "no reliable conclusions can be drawn from this analysis about the observed NO2 bias dependence on shadows". We also do not agree the we have a followed "questionable approach". The specific comments raised have been addressed in the answers below. In our opinion they do not change the main findings about the cloud shadow band cases discussed in 4.2.1 nor the results presented in the Appendix.

We agree that the results about the general cases in section 4.2.2 do not provide a strong case. Thus, we have removed the results from section 4.2.2 from the abstract and the conclusions. We still would argue that the data should be presented in order to guide future research on the topic.

We thus choose to keep section 4.2 with the changes described in the answers to the specific comments below.

**Detailed comments**

1. *Page 1, line 10-11: limit your conclusion to only synthetic data.*

   For the observational data we choose to keep the conclusion regarding the cloud shadow band. This is further discussed in comment 47 and 55 below. We have removed the conclusion concerning the data for general cases, see comment 56.

2. *Page 1, line 11-14: the low NO2 bias in the observations is not significant (see later comments below).*

   Please see answers to comments 47, 55 and 56 below.

3. *Page 4, Fig 1.b and Fig 1.c: Please avoid using a white color for the cloud pixels, since white is also one of the colors in the color bar.*

   The clouds are now white in both Fig 1b and 1c. The color map have been changed to avoid any confusion.

4. *Page 4, Fig 1.c: The cloud shadow index is 1 below the clouds, but this would be invisible from space. I suggest removing the cloud pixels in this plot (such as in Fig. 1.b), such that the cloud shadows become visible.*

   The color map has been changed and the cloudy pixels are shown as white, thus the cloud shadows are clearly visible.

5. *Page 5, Fig 2.a: It is difficult to see the colors of the RGB. Can you try to increase the brightness and/or enhance the colors?*

We have adjusted the lightness of the rgb plot so the colors are easier seen.

6. *Page 5: Fig 2: Please increase the size of the images. Discuss every subfigure when you introduce Fig. 2 in the text or remove the subfigure.*

To increase the image sizes would make the manuscript overly long. However, if the editors finds this useful the sizes may be increased. A brief discussion of the subfigures have been added to the text.

7. *Page 6, line 13: Why do you mention FRESCO here? Are you using FRESCO for your analysis? This is not yet clear for the reader at this point.*

It is mentioned on Page 3, line 30, that cloud corrections are made using the $O_2$A-band. We have added FRESCO in parenthesis at this point.

8. *Page 6, line 28: You do not consider ocean cases. On Page 3, line 18, you mention that your focus is on Europe and different cloud types, and that therefore the results are expected to be general and applicable elsewhere. Would your results also be representative over ocean? And over desert, or over snow/ice?*

In the revised conclusions we make the statement "Profiles of $NO_2$ for polluted conditions, with increased $NO_2$ in the lower atmosphere below cloud tops, were considered as cloud shadow effects are not important for background $NO_2$ conditions where the amount of $NO_2$ below the cloud top is relatively small compared to the total column." Most of the Earth's ocean may be considered representative for background conditions. We have not specifically looked at desert or snow/ice surfaces, but we can not see why the results should not be applicable to polluted conditions for such surfaces as well. In the accompanying manuscript by Yu et al. (2021) the albedo dependence of the retrieval error is discussed.

9. *Page 7, line 1-3: they compare the spectral procedure, and found the latter to be far superior. Could you rephrase this? What is the spectral procedure? What do you mean by far superior?*

We have rephrased the sentence to: "The VIIRS cloud shadow mask algorithm is geometry-based and described by Hutchison et al. (2009). They compared the MODIS MOD35 product, which uses spectral signatures to identify cloud shadows, with geometry-based approaches and states that the latter "are far superior to those predicted with the spectral procedures".

Note that the phrase "far superior" was used by Hutchison et al. (2009). This we have clarified by quoting Hutchison et al. (2009). Also note that Hutchison et al. (2009)provide no quantitative measure of the cloud shadow products, but rather make a qualitative comparison.

10. *Page 7: what are the accuracies of the VIIRS cloud mask and VIIRS shadow mask (regardless of the mapping to the TROPOMI grid)?*

The performance of the VIIRS cloud mask have been discussed by Hutchison et al. (2014). Over land they found agreement of 94.4% and 93.0% with manually generated cloud masks and CALIOP-VIIRS match-up datasets, respectively. The VIIRS cloud shadow mask is described by Hutchison et al. (2009). They do not provide a quantitative estimate of the VIIRS cloud shadow accuracy, but present convincing results of the performance of their algorithm. We are not aware of any other descriptions of such accuracy estimates and it is clearly outside the scope of this manuscript to provide such estimates.

11. *Page 7, line 16: From this explanation, it seems that the Cloud Shadow Index (CSI) also indicates fully cloudy pixels (as also shown in Fig. 1.c). Why is this the case? Wouldnt excluding cloudy pixels be better for the analysis of shadows which uses the CSI, such as Fig. 3? Or did you indeed apply a cloud filter? Please explain in this subsection, this is not clear for the reader at this point.*

    Fig. 1c has been redone and the CSI is no longer shown for the cloudy pixels. It is stated in the caption of Fig. 1 that the retrieval is not done for cloudy pixels.

12. *Page 7, line 29-30: Why are the " " used here? Is this a citation? This is not a proper explanation of the AAI. Please rephrase and add a reference to de Graaf et al. (2005):* `https://agupubs.onlinelibrary.wiley.com/doi/full/10.1029/2004JD005178`*. For example: The AAI is a measure of the UV color of a cloud-, aerosol- and shadow- free 1-D atmosphere-surface model with respect to the measured UV color (de Graaf et al., 2005). When absorbing aerosols are present, the AAI tends to be positive, while the AAI is approximately zero or negative in the presence of clouds (see e.g. Kooreman et al., 2020; Penning de Vries et al., 2009).*

    The " " indicates a quote and it is taken from Kooreman et al. (2020). We have rephrased the sentences as suggested.

13. *Page 8, Fig 3: Please remove cloud pixels from your analysis, and if you did (already), please mention this in the caption of the figure.*

    Cloud pixels are not included in the analysis. We have rephrased the caption to mention this.

14. *Page 8, line 14-16: The cloud shadow impact ... respectively. Please rephrase, this sentence is difficult to read. Are the percentages you mention here average values for a CSI of 1?*

    We have rephrased the sentence so it now reads "The cloud shadow impact is seen to increase as the solar zenith angle increases. The number of pixels with $NO_2$ TVCD differences $< -20\%$ is 0.1% for a solar zenith angle of 20° (Fig. 3a), 4.% for 40° (Fig. 3b) and 20.3% for 60° (Fig. 3c)."

    As stated in the text, the percentages are the number of pixels with $NO_2$ TVCD differences $< -20\%$.

15. *Page 8, line 16: As the solar zenith angle increases a linear relationship appears .... Is there really a linear relationship? The data looks scattered. Can you please quantify the linear relationship with the corresponding uncertainty?*

We have added linear fits to the data in Fig. 3. including $R^2$-values. The caption of Fig. 3 and the text has been changed accordingly.

16. *Page 8, line 20: For both geometries the NO2 AMF is high ... is between 1-3%. Why? Can you physically explain these numbers here already? If not, please refer to this location in the paper explicitly when you can. For example, on page 12 line 11, you investigated the cause, and you can refer back: This explains the high AMF biased that we observed in Fig. 4a for CFw < 1%.*

We have rearranged the text as suggested. Please also see answer to comment 25.

17. *Page 9, line 3: ... there are comparatively more pixels with a negative bias for LEO geometry. Why? In lines 4 to 14 you explain that this is because the SAA en SZA are different, giving different sensitivity to cloud shadows for LEO and GEO geometries. Can you explain why this is the case?*

We have added the following explanation:
"For the LEO and GEO geometries studied, see Emde et al. (2021) for details, the sun is to the south of the study region. This implies that a relatively large portion of cloud shadows are on the northern sides of the clouds. These cloud shadows are partly hidden from GEO satellites but may be visible from LEO satellite instrument with a nadir view of Earth, thus giving different sensitivity to cloud shadows for LEO and GEO geometries."

18. *Page 9, first paragraph: Please discuss Figure 5 in a separate paragraph.*

We have made a separate paragraph for the Fig. 5 discussion.

19. *Page 9 and 10, general comment: The results of the parameters such as SZA and surface albedo are discussed. The physical explanation is missing. After each finding, can make a connection here with the theory from your first paper (Emde et al. 2021)?*

When discussing Fig. 3 we have added an explanation of the solar zenith angle dependence and connected this with the paper of Emde et al. (2021). We have added further discussion, as suggested, in connection with the presentation of Fig. 4 and references are made to the papers by Emde et al. (2021) and Yu et al. (2021).

20. *Page 9 and 10, please explain better from theory of Emde et al. (2021) what the reader should be aware of when comparing LEO and GEO images (given the different SAA and SZA). What are the interesting differences between LEO and GEO results that you expect to see? And do you also observe in these simulated results what you expect from theory (Emde et al. 2021)?*

We have added a discussion about SSA and SZA differences between LEO and GEO geometries, see comment 17. Furthermore discussion with references to the papers by Emde et al. (2021) and Yu et al. (2021) have been added, see comments 17, 18, and 19.

21. *Page 10, Fig. 5: Why are the dots connected by lines, for example for (SZA=20 deg; SAA=45 deg) and (SZA = 40 deg, SAA = 270 deg)? Please reconsider the presentation of these results. Using 9 lines (for different albedo and SZA) instead of 3, or a bar chart, would suit better here.*

    Fig. 5 has been redone with 9 lines as suggested.

22. *Page 12, line 4: east/west. How is the solar azimuth precisely defined? Make clear which SAAs belong to west and east.*

    We have changed "west/east" to " west (SAA=-90°)/east (SAA=90°)" to indicate which SAA that belong to east and east.

23. *Page 12, Fig 8: please relocate the tick labels of the color bar such that it is clear to which color they belong.*

    The tick labels have been relocated.

24. *Page 12, Fig 8: What are the tick labels 40 and 20?*

    The labels have been changed so that the meaning is clear.

25. *Page 12, line 7-11: Generally ... effects. These lines are floating in the rest of the text, because they are a discussing of Fig. 4. Please move those lines to the discussion of Fig. 4, or make a connection to the former paragraphs.*

    We have adopted these lines and moved them to the discussion of Fig. 4 as suggested.

26. *Page 13-14, general comment: Please reconsider Figures 9 and 10. Consider replacing Figures 9 and 10 by figures that show the NO2 bias as functions of physical quantities such as SZA and albedo. This could make it easier to connect with the theory of Emde et al. (2021).*

    Figs. 9 and 10 have been replaced by one figure which shows the NO2 bias as functions of physical quantities.

27. *Page 13, Fig 9: Fig. 9 contains too much information. Why are the lines connected? A bar chart may suit better here. What are the different case numbers? It is not clear from the figure or the caption. Please prioritize the results you want to show and possibly compute averages of the cases. Think about the message you want to convey with this figure.*

    Figs. 9 and 10 have been replaced by one figure. The case numbers have been replaced by solar and viewing angles and the reason for connecting the lines given in the caption.

28. *Page 13, Fig 9b: Is Figure 9.b really needed for the conclusions of your paper? Similar comment for Fig. 10b.*

    Figs. 9b and 10b have been removed from the revised manuscript and the text adopted accordingly.

29. *Page 13, line 11: for similar reasons: What reasons? Additionally, are the reasons of the contamination as functions of SZA and VZA really (expected to be) identical? If yes, why?*

To clarify we have rewritten the sentence so it now reads: "This is due to geometry reasons which cause the cloud shadow to increase as the solar zenith angle increases. Also, as the viewing zenith angle increases a larger, potentially cloud shadow impacted, horizontal surface area will be viewed due to geometry reasons and thus the cloud shadow effect increase with increased viewing zenith angle."

30. *Page 13, line 11-12: under- and overestimates, under- and overestimates of what? The NO2 bias or AMF bias?*

It should read "under- and overestimates of the $NO_2$ TVCD". This has been corrected.

31. *Page 13, line 12: Cloud shadows are a cloud feature metric that may be used to identify affected pixels, Fig 3. What do you mean by this sentence?*

We have rewritten this sentence to "The underestimates are due to cloud shadows, thus the cloud shadow fraction is a cloud feature metric that may be used to identify affected pixels, Fig. 3."

32. *Page 14, line 1: occurr. -> occur.*

Corrected.

33. *Page 14, line 1: ... also occur. Can you refer to the figure(s) where this was shown?*

We have clarified this sentence so it now reads: "However, while for large solar zenith angles pixels affected by cloud shadows are mostly underestimated, overestimates occur for all solar zenith angles, is mostly present for low cloud shadow fractions (Fig. 3) and increase for large surface albedo (blue dashed lines Fig. 9)."

34. *Page 14, line 1-2: ..., such as cloud top altitude and cloud optical thickness, are also of importance. How did you come to this conclusion? Can you show this or refer to the figure where this has been shown?*

This is shown by Emde et al. (2021) and we have added this reference to the sentence.

35. *Page 14, line 5: How precisely is the cloud enhancement effect visible in Figs. 9 and 10? Please explain.*

Figs. 9 and 10 have been reworked as mentioned above. The cloud enhancement effect is seen in the blue lines in the revised figures and this is now mentioned in the text.

36. *Page 14, line 7: "theta = 20 – 30 degrees". How can the viewing zenith angle be observed in Fig. 9a?*

The solar and viewing angles are given in the x-tick labels in the revised Fig. 9.

37. *Page 14, line 12 to Page 15, line 4: What is the message of this paragraph? Do you mean that 3D cloud and cloud shadow effects are smaller than the NO2 retrieval uncertainty?*

    The purpose of this paragraph is to compare the magnitude of the 3D cloud error with other NO$_2$ retrieval errors. We have rewritten the paragraph to clarify this.

38. *Page 15, line 7-12: Should these sentences be part of Section 4.2.2 instead?*

    We have moved these sentences to Section 4.2.2 as suggested. The text in section 4.2.2 has been slightly adjusted to accomodate this move.

39. *Page 15, line 20-22: please rephrase: : For a cloud shadow fraction ... standard deviation. Please add: The scatter in Fig. 11.g is too large to draw conclusions about the dependence of NO2 on shadow fraction.*

    We have added a phrase at these lines as suggested.

40. *Page 15, line 24-25: Thus indicating that ... cloud cases. Please rephrase this sentence: what is the subject of this sentence?*

    We have rephrased this sentence.

41. *Page 16: can you please make the figures bigger?. Also, in Fig. 11.g, the lime green squares are not visible.*

    The size of Figs. 11b-11f have been increased. The size of the lime green squares in Fig. 11 has been increased to make them visible.

42. *Page 16, Fig. 11: can you explain the oscillatory pattern in the geometric cloud fraction (Fig. 11.d) and the cloud shadow fraction (Fig. 11.f) in the shadow band?*

    We have added the following text explaining this pattern: "The cloud shadow band has a width about the extent of 1-2 TROPOMI pixels. As the cloud shadow band and the TROPOMI pixels are not aligned this implies that the cloud shadow band at some locations will be completely covered by one TROPOMI pixel and at other locations partly covered by two TROPOMI pixels. This causes the oscillatory pattern seen in the geometric cloud fraction (Fig. 11.d) and the cloud shadow fraction (Fig. 11.f) in the cloud shadow band."

43. *Page 16, fig 11.g: how do you precisely define CSF pixels? Doesnt each pixel has a certain CSF? Please clarify this in the caption.*

    In the annotation of Fig .11g it should read "CSF> 0 pixels". This has been corrected.

44. *Page 16, Fig 11.g: the variability is much larger than the differences between NO2 (all pixels) and NO2 (CSF pixels). No significant relation between NO2 and CSF can be identified with this figure. Please clarify this explicitly in the text.*

    This has now been mentioned explictily in the text. See answer to comment 39 above.

45. *Page 17, line 4: For the cloud shadow band the NO2 TVCD is on average reduced by 17%. Please add a sentence here explaining that only a few pixel rows are analyzed, while the NO2 natural spatial variation is actually very large (Fig. 12.d).*

   To clarify this we have changed this sentence so it now reads: "While the $NO_2$ spatial varibility is large (Fig. 12.d), within the cloud shadow band covered by rows 262-269, the $NO_2$ TVCD is on average reduced by 17%."

46. *Page 17, line 22-23: All cases show that the NO2 TVCD in the cloud shadow is lower by 8-46% (average of 25%) compared with the NO2 TVCD around the shadow. What about pixel row 396? Pixel row 396 seems to have a higher NO2 TVCD in the shadow than south of shadow.*

   We clarified this by changing the sentence to: "With the exception of the cloudy pixels south of the cloud band for row 396, all other cases show that the $NO_2$ TVCD in the cloud shadow is lower by 8-46% (average of 25%) compared with the $NO_2$ TVCD around the shadow."

47. *Page 17, line 26-28: If it is assumed that the clouds are the main reason for the variations in the NO2 TVCD over the cloud shadow bands, then these cases are examples of how cloud shadows give underestimates of NO2 TVCD, in agreement with the theoretical idealized box cloud results presented by Emde et al. (2021) and Yu et al. (2021). I dont think you can conclude this, given the high spatial NO2 variability (Figs. 2.c, 12.d, 13.d), the limited number of cases and pixel rows that were analyzed, the high scatter of the NO2 bias as functions of shadow fraction (Fig. 11.g), and the questionable approach to mask clouds and shadows using VIIRS masks on the TROPOMI grid (due to the cloud movement and cloud evolution during the TROPOMI-VIIRS overpass time difference, the undiscussed VIIRS mask accuracy, and the oscillatory features in the geometric cloud fraction and shadow fraction in the shadow band (Fig.s 11d and 11f)).*

   Given the high spatial variability in the $NO_2$ TVCD we are actually surprised to find that for most cases the $NO_2$ TVCD is smaller in the cloud shadow band than outside it. The one explanation we have for this decrease in $NO_2$ TVCD is the 3D cloud effect presented by Emde et al. (2021). We thus find that despite the large spatial variability in the $NO_2$ TVCD, there is a clear signal of $NO_2$ TVCD decrease in the cloud shadow bands.

   Note that Fig. 11g includes all cloud shadow pixels in the image and not only the cloud band, thus the high scatter. This has been clarified in the text.

   The oscillatory features in the geometric cloud fraction and shadow fraction have been addressed in comment 42. The accuracies of the VIIRS cloud and cloud shadow masks have been addressed in comment 10.

   Concerning the movement of clouds between the S5P and S-NPP overpasses we have added the following text to the manuscript:

   "The time difference between the VIIRS and TROPOMI overpasses is about 4.2 min for the two cloud shadow band cases. For fast moving clouds this may give a shift in cloud and cloud shadow locations. For the two cloud shadow band cases discussed we investigated both ERA5 wind data and Spinning Enhanced Visible

and InfraRed Imager (SEVIRI) RGB images. The SEVIRI images have a time resolution of 15 min. and clearly show a southward movement of the cloud bands. The spatial resolution of SEVIRI together with possible cloud development make it challenging to precisely determine the speed of the cloud movement. We, however, estimate it to be on the order of 10-15 m/s in the southward direction perpendicular to the cloud shadow band. The ERA5 data have a large eastward component at the altitudes of the two cloud bands. For the 30 December 2019 case there is a much smaller southward component of about 10 m/s in agreement with the SEVIRI images. Surprisingly, for the 24 March 2019 case, the ERA5 data have a northward component of about 10 m/s, which is in disagreement with the SEVIRI observations. Trusting the SEVIRI images we find that the cloud mask and cloud shadow mask have shifted between 2.5 and 3.75 km perpendicular to the cloud shadow band between the TROPOMI and VIIRS overpasses. This is about the TROPOMI pixel size in this direction. For the 24 March 2019 case the cloud shadow band covers 1-2 TROPOMI pixels and it covers 2-4 TROPOMI pixels for the 30 December 2019 case. The cloud shadow band first viewed by VIIRS may thus be shifted southward when TROPOMI passes over. For the same geolocation, TROPOMI may thus view a smaller part of the cloud shadow band than VIIRS and hence be less affected by the cloud shadow. In Figs. 12 and 13 we average over the TROPOMI pixels identified to be affected by cloud shadow according to the VIIRS cloud shadow mask. Despite a possible reduction in the cloud shadow viewed by TROPOMI, a decrease is seen in the $NO_2$ TVCD for these pixels. We note that the cloud shift may in principle be corrected for using for example ERA5 data. However, as reported above, we find that SEVIRI and ERA5 data give different results."

48. *Page 17, line 34: TROPOMI processes 25 million pixels per day. Why do you use for October 2018 and March 2019 only 1023081 pixels? What is the study region precisely?*

   The study region is described in the Introduction. To clarify this we added the following text repeating the study region description "(covering approximately Germany, the Netherlands and parts of other surrounding countries, see Introduction)". The reason for using the months of October 2018 and March 2019 is the solar zenith angle as explained in the text.

49. *Page 18, line 1: 35% of what precisely? 35% is a large percentage for cloud shadows, even in months where you expect cloud shadows. Can you please verify this number? How does this number relate to the overall cloud fraction of the data set?*

   We have clarified this sentence and related it to the overall cloudiness as follows:
   "A $NO_2$ retrieval with the data quality value $>0.95$ was reported for 367,584 (36%) of the pixels. The VIIRS cloud mask identified 70.7% of the VIIRS pixels to be cloudy, indicating that clouds were the main reason for reducing the $NO_2$ retrieval quality for the majority of the TROPOMI pixels. Of the 367,584 pixels with high $NO_2$ retrieval data quality, a total of 129,180 (35%) were affected by cloud shadows according to the VIIRS cloud shadow product. Of the 45,926,808 VIIRS pixels 1,3438,968 (29.3%) were cloud free. Of these cloud free VIIRS pixels 17.8% contained cloud

shadows. This number is lower than the number of TROPOMI pixels affect by cloud shadows as is to be expected due to the higher spatial resolution of VIIRS."

50. *Page 18 and 19, Fig 12 and 13: Fig 12 and 13 are hard to read. Please make the figures bigger. Figure 12.b and 13.b are problematic: can you ensure that the cloud movement and evolution during the TROPOMI-VIIRS overpass time difference did not consistently affect the shadow identification?*

We have enlarged Fig 12 and 13 by about 30%. For cloud movement discussion please see answer to comment 47.

51. *Page 18 and page 19, Fig.12.g and 13.g: only a couple of pixel rows are analyzed, and even within this small sample, the low NO2 bias is not consistent. For example, in Fig. 12.g, the NO2 TVCD is higher in the shadow than outside the shadow for rows 262 and 265.*

We presume it is rows 262 and 269 that are meant. For these two rows the NO2 TVCD is smaller in the cloudless regions to the north of the cloud band compared to the shadow region. In the paper we discuss the problem of not having a "true" $NO_2$ TVCD. Thus, as clearly stated in the manuscript, our conclusions about the cloud shadow bands are based on the assumption that the $NO_2$ field is horizontally homogeneous. That this assumption may not hold for all cases is to be expected. However, in the lack of a "true" $NO_2$ TVCD, this assumption appears to be a good first guess. We have modified the discussion of Fig. 12g as follows:

"Except for rows 262 and 269, the $NO_2$ TVCD is smaller in the cloud shadow band compared to the $NO_2$ TVCD north of the cloud shadow. The $NO_2$ spatial varibility is large (Fig. 12.d), despite this, for the cloud shadow band covered by rows 262-269, the $NO_2$ TVCD is on average reduced by 17%."

For Fig. 13g the $NO_2$ TVCD is lower in the cloud shadow band for all cases presented.

52. *Page 20, line 10: no true NO2 TVCD is available as for the synthetic data -> do you mean observational data?*

Yes, no true observational $NO_2$ TVCD is available. This has been clarified in the text.

53. *Section 4.2.2 general comment: The results in Figs. 14 and 15 are highly scattered, and no clear negative NO2 bias from cloud shadows can be determined. This should be clear in the text, conclusions and abstract.*

It is written in section 4.2.2 that "no signifcant cloud shadow effect is visible in the $NO_2$ TVCD" for the data presented in Fig. 14 (Fig. 13 in revised manuscript). The data presented in Fig. 15 (Fig. 14 in revised manuscript) is carefully discussed in secrtion 4.2.2 without making any firm conclusions due to the uncertainty in the data.

We have removed the reference to these data in the abstract and conclusions, see comments 1 and 56.

54. *Section 4.2.2 general comment: neighbor pixels in a 3x3 pixel matrix where used, and the true NO2 TVCD is then taken to be the average of the cloud-free neighbors. Cloud movement and evolution during the measurements time difference of TROPMI and VIIRS could consistently result in the situation where the actual shadows are located inside the neighboring pixels, while the identified shadow pixels are in fact shadow- free.*

We have added a sentence mentioning the possiblity for cloud movement between VIIRS and TROPOMI overpasses.

55. *Page 23, line 15-19: For clearly identified cloud shadow bands ... with the theoretical findings. Why can you assume that the clouds are the main reason for the spatial NO2 variations / assume that the NO2 background is horizontally homogeneous?*

We discuss this in the answer to comment 47.

56. *Page 23, line 20-21: For a solar zenith ... to be impacted by cloud effects larger than 20%. Where did you show this? Also, please mention that the data is very scattered and comment on the uncertainty of your conclusions.*

This is shown at the end of section 4.2.2. Due to the high uncertainty in these numbers we have omitted them from the abstract and the conclusions.

57. *Page 24, line 1: You mention that there are large changes between versions of the VIIRS cloud shadow product. Could you elaborate on that? What is the accuracy of the VIIRS cloud shadow product itself (regardless of the mapping onto the TROPOMI grid)?*

We have added a footnote with the following text "The VIIRS L2 product changed version from v1r1 to v1r2 between 13 and 14 Aug 2018, see `https://www.star.nesdis.noaa.gov/jpss/documents/AMM/N20/Cloud_CBH_Provisional.pdf`. Large changes in the cloud shadow product was seen between versions with v1r1 given unrealistic large number of pixels with cloud shadow. Realistic numbers were found with v1r2."

Concerning the accuracy of the VIIRS cloud shadow product please see answer to comment 10.

58. *Page 24, line 12: As cloud shadow impact NO2 TVCD retrievals, ... Do you mean instead: As cloud shadow impact both AAI and NO2 retrievals, ...?*

Change made as suggested.

59. *Page 24, line 15: Indeed, over land the AAI is more negative over cloudy pixels, compare Fig. 11d and Fig. A1a. -> This seems not really to be the case when looking at Fig. 11d and Fig. A1a: the large cloud deck between 52 deg N and 52.5 deg N does not give more negative AAI. Clouds do not always decrease the AAI, they usually just dont increase the AAI (see e.g. Penning de Vries et al., 2009).*

We have changed the quoted sentence to "The behaviour of clouds on AAI is complex. For effective cloud fraction between 30-50% (5-30%) for thick (thin) clouds

Penning de Vries et al. (2009) reported negative AAI while for large cloud fractions high, thick clouds may cause positive AAI. The increase in AAI from scattered clouds to complete cloud cover may be seen when comparing Fig. 11d and Fig. A1a." In addition we have changed the color scale of Fig. A1a to better visualize the AAI.

60. *Page 25, line 2: ..., while the NO2 TVCD shows some dependency on cloud shadow fraction, Fig. 11g. -> Please remove this part, the dependency on cloud shadow fraction from Fig. 11g is insignificant given the high variability.*

This part has been removed as suggested.

**Bibliography**

Emde, C., Yu, H., Emde, C., Kylling, A., van Roozendael, M., Stebel, K., Veihelmann, B., and Mayer, B.: Impact of 3D Cloud Structures on the Atmospheric Trace Gas Products from UV-VIS Sounders - Part I: Synthetic dataset for validation of trace gas retrieval algorithms, Atmospheric Measurement Techniques, submitted, 2021.

Hutchison, K. D., Mahoney, R. L., Vermote, E. F., Kopp, T. J., Jackson, J. M., Sei, A., and Iisager, B. D.: A Geometry-Based Approach to Identifying Cloud Shadows in the VIIRS Cloud Mask Algorithm for NPOESS, Journal of Atmospheric and Oceanic Technology, 26, 1388–1397, https://doi.org/10.1175/2009JTECHA1198.1, URL `https://doi.org/10.1175/2009JTECHA1198.1`, 2009.

Hutchison, K. D., Heidinger, A. K., Kopp, T. J., Iisager, B. D., and Frey, R. A.: Comparisons between VIIRS cloud mask performance results from manually generated cloud masks of VIIRS imagery and CALIOP-VIIRS match-ups, International Journal of Remote Sensing, 35, 4905–4922, https://doi.org/10.1080/01431161.2014.932465, URL `https://doi.org/10.1080/01431161.2014.932465`, 2014.

Kooreman, M. L., Stammes, P., Trees, V., Sneep, M., Tilstra, L. G., de Graaf, M., Stein Zweers, D. C., Wang, P., Tuinder, O. N. E., and Veefkind, J. P.: Effects of clouds on the UV Absorbing Aerosol Index from TROPOMI, Atmospheric Measurement Techniques Discussions, 2020, 1–31, https://doi.org/10.5194/amt-2020-112, URL `https://www.atmos-meas-tech-discuss.net/amt-2020-112/`, 2020.

Penning de Vries, M. J. M., Beirle, S., and Wagner, T.: UV Aerosol Indices from SCIAMACHY: introducing the SCattering Index (SCI), Atmospheric Chemistry and Physics, 9, 9555–9567, https://doi.org/10.5194/acp-9-9555-2009, URL `https://acp.copernicus.org/articles/9/9555/2009/`, 2009.

Yu, H., Emde, C., Kylling, A., Veihelmann, B., Mayer, B., Stebel, K., and van Roozendael, M.: Impact of 3D Cloud Structures on the Atmospheric Trace Gas Products from UV-VIS Sounders - Part II: impact on NO 2 retrieval and mitigation strategies, Atmospheric Measurement Techniques, submitted, 2021.

---

## Author Response (AR2)

**Response to comments from editor**

Please find below our response (in roman font) to your comments (in italic font).

**General comments**

- *I tend to agree with Rev#2 casting some doubt on whether the observational data clearly show a low bias on the order of tens of % in the NO2 vertical column. In the end, your Figure 14 seems to point in that way, but this mainly appears as an indirect, statistical indication. The evidence from the synthetic data is much stronger. The manuscript needs to (re-)written such, that the reader is clearly guided that the 'observational evidence' is much more indirect and weaker than the synthetic argument. For example on page 1, line 11 and page 8, L11-12 it is mentioned that the low bias in tropospheric NO2 can de derived from observational data. I think this statement should be toned down.*

  We have toned down the observational data in the Abstract and in the Conclusions. In the Abstract (a similar change has been made in the Conclusions) the last three sentences have been replaced by the following:

  > For a solar zenith angle less than about $40°$ the synthetic data show that the $NO_2$ TVCD bias is typically below 10%, while for larger solar zenith angles the $NO_2$ TVCD is low-biased by tens of %. The horizontal variability of $NO_2$ and differences in TROPOMI and VIIRS overpass times makes it challenging to identify a similar bias in the observational data. However, for optically thick clouds above 3000 m a low bias appears to be present in the observational data.

  On page 8, L11-L12 the text now reads.

  > For the observational data the true $NO_2$ TVCD unaffected by clouds is in general not known and is difficult to estimate due to the horizontal variability of $NO_2$. An attempt to estimate the true $NO_2$ TVCD from the observational data is discussed in section 4.2, which also include the analysis of the observational data.

- *Please consider moving the subsection on cloud movement to the Supplementary Material*

  Please see answer below to comment about section 4.2 and Figs. 10 and 11.

- *I find Figure 9 quite difficult to interpret. That SZA is an important driver of cloud-shadow induced errors has been made clear already before. Please consider removing this figure, or moving it to the supplement, and shortening the discussion.*

  We have moved Fig. 9 and the discussion of it to the supplement.

- *In section 4.2 Figures 10 and 11 are merely telling us that VIIRS data can be useful to inform us about cloud properties relevant to shadow working, but the figure does not show a strong relationship between shadow indicators with TROPOMI NO2 levels. Indeed spatial variability in NO2 is stronger than the shadow effect. There may still be good reasons to keep the figure (which in my opinion is the basis for Figure 14), but refer to it in a supplement rather than in the main manuscript, where it takes quite some mental space from the reader to follow what you are driving at, which seems to be the statistical indications for a low bias effect shown in Figure 14.*

  We have moved Figs. 10-12 to the supplement. As the discussion of these Figures are full of references to them, we have also moved the discussion the supplement. We have also moved the appendix to the supplement as it is coupled to the cloud shadow band cases.

- P21, L23-27: please make clear here that the low bias is found specifically for clouds with high cloud optical thickness.

  This has been made clear.

- *Overall, please carefully read the entire manuscript and assess how the text can be made more concise. The current manuscript is quite lengthy and technical, and it should be possible to reduce the text and material by at least 10%. This will improve the readability of the paper, and guide readers to the main take-home messages of the study.*

  We have carefully read the manuscript. The movement of material to the supplement has shortened the main text by about 25%. Furthermore, text in the introduction that was duplicated in section 2 has been removed.

**Minor comments**

- *In some places you mention 'TROPOMO' instead of TROPOMI.*

  Corrected.

- *4$\dot{\%}$ → 4% or 4.0%*

  Changed to 4.1%. Also 0.1% changed to 0.2% and 20.3% changed to 20.1% in the same sentence so numbers agree with those reported in Fig. 3.

- *Page 3, line 2: please provide a reference to the operational TROPOMI NO2 data product, i.e. van Geffen, J.H.G.M., Eskes, H.J., Boersma, K.F., Maasakkers, J.D. and Veefkind, J.P., TROPOMI ATBD of the total and tropospheric NO2 data products, Report S5P-KNMI-L2-0005-RP, KNMI, De Bilt, The Netherlands.*

  Reference included as suggested.

- *Page 3, L4-5: I suggest to introduce the small differences in overpass time here.*

We have added the sentence below. However, as the introduction as been shortened, this sentence comes in a later section.

> The difference in overpass time is slightly more than four minutes and care must be taken to for example movement of clouds when combining data from the two platforms (e.g. Trees et al., 2021).

- *P21: please make clear what the source of information is for the SCOT.*

  The following explanation has been added:

  $SCOT = COT/\cos(\theta)$, where COT is the VIIRS cloud optical thickness.

**Bibliography**

Trees, V., Wang, P., Stammes, P., Tilstra, L. G., Donovan, D. P., and Siebesma, A. P.:
DARCLOS: a cloud shadow detection algorithm for TROPOMI, Atmospheric Measurement Techniques Discussions, 2021, 1–29, https://doi.org/10.5194/amt-2021-377, URL
https://amt.copernicus.org/preprints/amt-2021-377/, 2021.

---

## Author Response (AR3)

**Response to comments from editor**

Please find below our response (in roman font) to your comments (in italic font). We have also made some clarifications to the Supplement.

**Minor comments**

- *P3, L12: please indicate the spatial resolution as 7 km x 7 km or 7 x 7 km² here.*
  Corrected to 7×7 km².

- *P6, L10: I suggest to indicate the wavelength here as 0.488 um or 488 nm.*
  Corrected to 0.488 μm.

- *P8, L6: 'include' –> includes*
  Corrected.

- *P14, L33: 'signifcant' -– > significant*
  Corrected.

- P17, L28: 'total' –> tropospheric. Earlier you mentioned that no stratospheric NO2 amounts are simulated.
  Corrected.